# TOWARDS ROBUST, LOCALLY LINEAR DEEP NETWORKS

**Guang-He Lee, David Alvarez-Melis & Tommi S. Jaakkola**
Computer Science and Artificial Intelligence Lab
MIT
{guanghe,davidam,tommi}@csail.mit.edu

## ABSTRACT

Deep networks realize complex mappings that are often understood by their locally linear behavior at or around points of interest. For example, we use the derivative of the mapping with respect to its inputs for sensitivity analysis, or to explain (obtain coordinate relevance for) a prediction. One key challenge is that such derivatives are themselves inherently unstable. In this paper, we propose a new learning problem to encourage deep networks to have stable derivatives over larger regions. While the problem is challenging in general, we focus on networks with piecewise linear activation functions. Our algorithm consists of an inference step that identifies a region around a point where linear approximation is provably stable, and an optimization step to expand such regions. We propose a novel relaxation to scale the algorithm to realistic models. We illustrate our method with residual and recurrent networks on image and sequence datasets.[1]

## 1 INTRODUCTION

Complex mappings are often characterized by their derivatives at points of interest. Such derivatives with respect to the inputs play key roles across many learning problems, including sensitivity analysis. The associated local linearization is frequently used to obtain explanations for model predictions (Baehrens et al., 2010; Simonyan et al., 2013; Sundararajan et al., 2017; Smilkov et al., 2017); explicit first-order local approximations (Rifai et al., 2012; Goodfellow et al., 2015; Wang & Liu, 2016; Koh & Liang, 2017; Alvarez-Melis & Jaakkola, 2018b); or used to guide learning through regularization of functional classes controlled by derivatives (Gulrajani et al., 2017; Bellemare et al., 2017; Mroueh et al., 2018). We emphasize that the derivatives discussed in this paper are with respect to the input coordinates rather than parameters.

The key challenge lies in the fact that derivatives of functions parameterized by deep learning models are not stable in general (Ghorbani et al., 2019). State-of-the-art deep learning models (He et al., 2016; Huang et al., 2017) are typically over-parametrized (Zhang et al., 2017), leading to unstable functions as a by-product. The instability is reflected in both the function values (Goodfellow et al., 2015) as well as the derivatives (Ghorbani et al., 2019; Alvarez-Melis & Jaakkola, 2018a). Due to unstable derivatives, first-order approximations used for explanations therefore also lack robustness (Ghorbani et al., 2019; Alvarez-Melis & Jaakkola, 2018a).

We note that gradient stability is a notion different from adversarial examples. A stable gradient can be large or small, so long as it remains approximately invariant within a local region. Adversarial examples, on the other hand, are small perturbations of the input that change the predicted output (Goodfellow et al., 2015). A large local gradient, whether stable or not in our sense, is likely to contribute to finding an adversarial example. Robust estimation techniques used to protect against adversarial examples (e.g., (Madry et al., 2018)) focus on stable function values rather than stable gradients but can nevertheless indirectly impact (potentially help) gradient stability. A direct extension of robust estimation to ensure gradient stability would involve finding maximally distorted derivatives and require access to approximate Hessians of deep networks.

---

[1]Project page: http://people.csail.mit.edu/guanghe/locally_linear.

In this paper, we focus on deep networks with piecewise linear activations to make the problem tractable. The special structure of this class of networks (functional characteristics) allows us to infer lower bounds on the $\ell_p$ *margin* — the maximum radius of $\ell_p$-norm balls around a point where derivatives are provably stable. In particular, we investigate the special case of $p = 2$ since the lower bound has an analytical solution, and permits us to formulate a regularization problem to maximize it. The resulting objective is, however, rigid and non-smooth, and we further relax the learning problem in a manner resembling (locally) support vector machines (SVM) (Vapnik, 1995; Cortes & Vapnik, 1995).

Both the inference and learning problems in our setting require evaluating the gradient of each neuron with respect to the inputs, which poses a significant computational challenge. For piecewise linear networks, given $D$-dimensional data, we propose a novel perturbation algorithm that collects all the *exact* gradients by means of forward propagating $O(D)$ carefully crafted samples *in parallel* without any back-propagation. When the GPU memory cannot fit $O(D)$ samples in one batch, we develop an unbiased approximation to the objective with a random subset of such samples.

Empirically, we examine our inference and learning algorithms with fully-connected (FC), residual (ResNet) (He et al., 2016), and recurrent (RNN) networks on image and time-series datasets with quantitative and qualitative experiments. The main contributions of this work are as follows:

- Inference algorithms that identify input regions of neural networks, with piecewise linear activation functions, that are provably stable.
- A novel learning criterion that effectively expand regions of provably stable derivatives.
- Novel perturbation algorithms that scale computation to high dimensional data.
- Empirical evaluation with several types of networks.

## 2 RELATED WORK

For tractability reasons, we focus in this paper on neural networks with piecewise linear activation functions, such as ReLU (Glorot et al., 2011) and its variants (Maas et al., 2013; He et al., 2015; Arjovsky et al., 2016). Since the nonlinear behavior of deep models is mostly governed by the activation function, a neural network defined with affine transformations and piecewise linear activation functions is inherently piecewise linear (Montufar et al., 2014). For example, FC, convolutional neural networks (CNN) (LeCun et al., 1998), RNN, and ResNet (He et al., 2016) are all plausible candidates under our consideration. We will call this kind of networks *piecewise linear networks* throughout the paper.

The proposed approach is based on a mixed integer linear representation of piecewise linear networks, *activation pattern* (Raghu et al., 2017), which encodes the active linear piece (integer) of the activation function for each neuron; once an activation pattern is fixed, the network degenerates to a linear model (linear). Thus the feasible set corresponding to an activation pattern in the input space is a natural region where derivatives are provably stable (same linear function). Note the possible degenerate case where neighboring regions (with different activation patterns) nevertheless have the same end-to-end linear coefficients (Serra et al., 2018). We call the feasible set induced by an activation pattern (Serra et al., 2018) a *linear region*, and a maximal connected subset of the input space subject to the same derivatives of the network (Montufar et al., 2014) a *complete linear region*. Activation pattern has been studied in various contexts, such as visualizing neurons (Fischetti & Jo, 2017), reachability of a specific output value (Lomuscio & Maganti, 2017), its connection to vector quantization (Balestriero & Baraniuk, 2018), counting the number of linear regions of piecewise linear networks (Raghu et al., 2017; Montúfar, 2017; Serra et al., 2018), and adversarial attacks (Cheng et al., 2017; Fischetti & Jo, 2017; Weng et al., 2018) or defense (Wong & Kolter, 2018). Note the distinction between locally linear regions of the functional mapping and decision regions defined by classes (Wong & Kolter, 2018; Yan et al., 2018; Mirman et al., 2018; Croce et al., 2019).

Here we elaborate differences between our work and the two most relevant categories above. In contrast to quantifying the number of linear regions as a measure of complexity, we focus on the local linear regions, and try to expand them via learning. The notion of stability we consider differs from adversarial examples. The methods themselves are also different. Finding the exact adversarial example is in general NP-complete (Katz et al., 2017; Sinha et al., 2018), and mixed integer linear programs that compute the exact adversarial example do not scale (Cheng et al., 2017; Fischetti & Jo, 2017). Layer-wise relaxations of ReLU activations (Weng et al., 2018; Wong & Kolter, 2018)

are more scalable but yield bounds instead exact solutions. Empirically, even relying on relaxations, the defense (learning) methods (Wong & Kolter, 2018; Wong et al., 2018) are still intractable on ImageNet scale images (Deng et al., 2009). In contrast, our inference algorithm certifies the exact $\ell_2$ margin around a point subject to its activation pattern by forwarding $O(D)$ samples in parallel. In a high-dimensional setting, where it is computationally challenging to compute the learning objective, we develop an unbiased estimation by a simple sub-sampling procedure, which scales to ResNet (He et al., 2016) on $299 \times 299 \times 3$ dimensional images in practice.

The proposed learning algorithm is based on the inference problem with $\ell_2$ margins. The derivation is reminiscent of the SVM objective (Vapnik, 1995; Cortes & Vapnik, 1995), but differs in its purpose; while SVM training seeks to maximize the $\ell_2$ margin between data points and a linear classifier, our approach instead maximizes the $\ell_2$ margin of linear regions around each data point. Since there is no label information to guide the learning algorithm for each linear region, the objective is unsupervised and more akin to transductive/semi-supervised SVM (TSVM) (Vapnik & Sterin, 1977; Bennett & Demiriz, 1999). In the literature, the idea of margin is also extended to nonlinear classifiers in terms of decision boundaries (Elsayed et al., 2018). Concurrently, Croce et al. (2019) also leverages the (raw) $\ell_p$ margin on small networks for adversarial training. In contrast, we develop a smooth relaxation of the $\ell_p$ margin and novel perturbation algorithms, which scale the computation to realistic networks, for gradient stability.

The problem we tackle has implications for interpretability and transparency of complex models. The gradient has been a building block for various explanation methods for deep models, including gradient saliency map (Simonyan et al., 2013) and its variants (Springenberg et al., 2014; Sundararajan et al., 2017; Smilkov et al., 2017), which apply a gradient-based attribution of the prediction to the input with nonlinear post-processings for visualization (e.g., normalizing and clipping by the $99^{\text{th}}$ percentile (Smilkov et al., 2017; Sundararajan et al., 2017)). While one of the motivations for this work is the instability of gradient-based explanations (Ghorbani et al., 2019; Alvarez-Melis & Jaakkola, 2018a), we focus more generally on the fundamental problem of establishing robust derivatives.

## 3 METHODOLOGY

To simplify the exposition, the approaches are developed under the notation of FC networks with ReLU activations, which naturally generalizes to other settings. We first introduce notation, and then present our inference and learning algorithms. All the proofs are provided in Appendix A.

### 3.1 NOTATION

We consider a neural network $\theta$ with $M$ hidden layers and $N_i$ neurons in the $i^{\text{th}}$ layer, and the corresponding function $f_\theta : \mathbb{R}^D \to \mathbb{R}^L$ it represents. We use $\mathbf{z}^i \in \mathbb{R}^{N_i}$ and $\mathbf{a}^i \in \mathbb{R}^{N_i}$ to denote the vector of (raw) neurons and activated neurons in the $i^{\text{th}}$ layer, respectively. We will use $\mathbf{x}$ and $\mathbf{a}^0$ interchangeably to represent an input instance from $\mathbb{R}^D = \mathbb{R}^{N_0}$. With an FC architecture and ReLU activations, each $\mathbf{a}^i$ and $\mathbf{z}^i$ are computed with the transformation matrix $\mathbf{W}^i \in \mathbb{R}^{N_i \times N_{i-1}}$ and bias vector $\mathbf{b}^i \in \mathbb{R}^{N_i}$ as

$$\mathbf{a}^i = \text{ReLU}(\mathbf{z}^i) := \max(\mathbf{0}, \mathbf{z}^i), \quad \mathbf{z}^i = \mathbf{W}^i \mathbf{a}^{i-1} + \mathbf{b}^i, \forall i \in [M], \quad \mathbf{a}^0 = \mathbf{x}, \tag{1}$$

where $[M]$ denotes the set $\{1, \dots, M\}$. We use subscript to further denote a specific neuron. To avoid confusion from other instances $\bar{\mathbf{x}} \in \mathbb{R}^D$, we assert all the neurons $\mathbf{z}_j^i$ are functions of the specific instance denoted by $\mathbf{x}$. The output of the network is a linear transformation of the last hidden layer $f_\theta(\mathbf{x}) = \mathbf{W}^{M+1} \mathbf{a}^M + \mathbf{b}^{M+1}$ with $\mathbf{W}^{M+1} \in \mathbb{R}^{L \times N_M}$ and $\mathbf{b}^{M+1} \in \mathbb{R}^L$. The output can be further processed by a nonlinearity such as softmax for classification problems. However, we focus on the piecewise linear property of neural networks represented by $f_\theta(\mathbf{x})$, and leverage a generic loss function $\mathcal{L}(f_\theta(\mathbf{x}), \mathbf{y})$ to fold such nonlinear mechanism.

We use $\mathcal{D}$ to denote the set of training data $(\mathbf{x}, \mathbf{y})$, $\mathcal{D}_{\mathbf{x}}$ to denote the same set without labels $\mathbf{y}$, and $\mathcal{B}_{\epsilon, p}(\mathbf{x}) := \{\bar{\mathbf{x}} \in \mathbb{R}^D : \|\bar{\mathbf{x}} - \mathbf{x}\|_p \leq \epsilon\}$ to denote the $\ell_p$-ball around $\mathbf{x}$ with radius $\epsilon$.

The activation pattern (Raghu et al., 2017) used in this paper is defined as:

**Definition 1.** *(Activation Pattern) An activation pattern is a set of indicators for neurons* $\mathcal{O} = \{\mathbf{o}^i \in \{-1, 1\}^{N_i} | i \in [M]\}$ *that specifies the following functional constraints:*

$$\mathbf{z}_j^i \geq 0, \quad \text{if} \quad \mathbf{o}_j^i = 1; \quad \mathbf{z}_j^i \leq 0, \quad \text{if} \quad \mathbf{o}_j^i = -1. \tag{2}$$

Each $\mathbf{o}_j^i$ is called an activation indicator. Note that a point on the boundary of a linear region is feasible for multiple activation patterns. The definition fits the property of the activation pattern discussed in §2. We define $\nabla_{\mathbf{x}}\mathbf{z}_j^i$ to be the sub-gradient found by back-propagation using $\partial\mathbf{a}_{j'}^{i'}/\partial\mathbf{z}_{j'}^{i'} := \max(\mathbf{o}_{j'}^{i'}, 0), \forall j' \in [N_{i'}], i' \in [i-1]$, whenever $\mathbf{o}_{j'}^{i'}$ is defined in the context.

## 3.2 Inference for Regions with Stable Derivatives

Although the activation pattern implicitly describes a linear region, it does not yield explicit constraints on the input space, making it hard to develop algorithms directly. Hence, we first derive an explicit characterization of the feasible set on the input space $\mathbb{R}^D$ with Lemma 2.[2]

**Lemma 2.** *Given an activation pattern $\mathcal{O}$ with any feasible point $\mathbf{x}$, each activation indicator $\mathbf{o}_j^i \in \mathcal{O}$ induces a feasible set $S_j^i(\mathbf{x}) = \{\bar{\mathbf{x}} \in \mathbb{R}^D : \mathbf{o}_j^i[(\nabla_{\mathbf{x}}\mathbf{z}_j^i)^\top\bar{\mathbf{x}} + (\mathbf{z}_j^i - (\nabla_{\mathbf{x}}\mathbf{z}_j^i)^\top\mathbf{x})] \geq 0\}$, and the feasible set of the activation pattern is equivalent to $S(\mathbf{x}) = \cap_{i=1}^M \cap_{j=1}^{N_i} S_j^i(\mathbf{x})$.*

**Remark 3.** *Lemma 2 characterizes each linear region of $f_\theta$ as the feasible set $S(\mathbf{x})$ with a set of linear constraints with respect to the input space $\mathbb{R}^D$, and thus $S(\mathbf{x})$ is a convex polyhedron.*

The aforementioned linear property of an activation pattern equipped with the input space constraints from Lemma 2 yield the definition of $\hat{\epsilon}_{\mathbf{x},p}$, the $\ell_p$ margin of $\mathbf{x}$ subject to its activation pattern:

$$\hat{\epsilon}_{\mathbf{x},p} := \max_{\epsilon \geq 0: \mathbf{x}' \in S(\mathbf{x}), \forall \mathbf{x}' \in \mathcal{B}_{\epsilon,p}(\mathbf{x})} \epsilon = \max_{\epsilon \geq 0: \mathcal{B}_{\epsilon,p}(\mathbf{x}) \subseteq S(\mathbf{x})} \epsilon, \tag{3}$$

where $S(\mathbf{x})$ can be based on *any* feasible activation pattern $\mathcal{O}$ on $\mathbf{x}$;[3] therefore, $\partial\mathbf{a}_j^i/\partial\mathbf{z}_j^i$ at $\mathbf{z}_j^i = 0$ from now on can take 0 or 1 arbitrarily as long as consistency among sub-gradients $\{\nabla_{\mathbf{x}}\mathbf{z}_j^i | j \in [N_i], i \in [M]\}$ is ensured with respect to some feasible activation pattern $\mathcal{O}$. Note that $\hat{\epsilon}_{\mathbf{x},p}$ is a lower bound of the $\ell_p$ margin subject to a derivative specification (i.e., a complete linear region).

### 3.2.1 Directional Verification, the Cases $p = 1$ and $p = \infty$

We first exploit the convexity of $S(\mathbf{x})$ to check the feasibility of a directional perturbation.

**Proposition 4.** *(Directional Feasibility) Given a point $\mathbf{x}$, a feasible set $S(\mathbf{x})$ and a unit vector $\Delta\mathbf{x}$, if $\exists \bar{\epsilon} \geq 0$ such that $\mathbf{x} + \bar{\epsilon}\Delta\mathbf{x} \in S(\mathbf{x})$, then $f_\theta$ is linear in $\{\mathbf{x} + \epsilon\Delta\mathbf{x} : 0 \leq \epsilon \leq \bar{\epsilon}\}$.*

The feasibility of $\mathbf{x} + \bar{\epsilon}\Delta\mathbf{x} \in S(\mathbf{x})$ can be computed by simply checking whether $\mathbf{x} + \epsilon\Delta\mathbf{x}$ satisfies the activation pattern $\mathcal{O}$ in $S(\mathbf{x})$. Proposition 4 can be applied to the feasibility problem on $\ell_1$-balls.

**Proposition 5.** *($\ell_1$-ball Feasibility) Given a point $\mathbf{x}$, a feasible set $S(\mathbf{x})$, and an $\ell_1$-ball $\mathcal{B}_{\epsilon,1}(\mathbf{x})$ with extreme points $\mathbf{x}^1, \ldots, \mathbf{x}^{2D}$, if $\mathbf{x}^i \in S(\mathbf{x}), \forall i \in [2D]$, then $f_\theta$ is linear in $\mathcal{B}_{\epsilon,1}(\mathbf{x})$.*

Proposition 5 can be generalized for an $\ell_\infty$-ball. However, in high dimension $D$, the number of extreme points of an $\ell_\infty$-ball is exponential to $D$, making it intractable. Instead, the number of extreme points of an $\ell_1$-ball is only linear to $D$ ($+\epsilon$ and $-\epsilon$ for each dimension). With the above methods to verify feasibility, we can do binary searches to find the certificates of the margins for directional perturbations $\hat{\epsilon}_{\mathbf{x},\Delta\mathbf{x}} := \max_{\{\epsilon \geq 0: \mathbf{x} + \epsilon\Delta\mathbf{x} \in S(\mathbf{x})\}} \epsilon$ and $\ell_1$-balls $\hat{\epsilon}_{\mathbf{x},1}$. The details are in Appendix B.

### 3.2.2 The case $p = 2$

The feasibility on $\hat{\epsilon}_{\mathbf{x},1}$ is tractable due to convexity of $S(\mathbf{x})$ and its certification is efficient by a binary search; by further exploiting the polyhedron structure of $S(\mathbf{x})$, $\hat{\epsilon}_{\mathbf{x},2}$ can be certified analytically.

**Proposition 6.** *($\ell_2$-ball Certificate) Given a point $\mathbf{x}$, $\hat{\epsilon}_{\mathbf{x},2}$ is the minimum $\ell_2$ distance between $\mathbf{x}$ and the union of hyperplanes $\cup_{i=1}^M \cup_{j=1}^{N_i} \{\bar{\mathbf{x}} \in \mathbb{R}^D : (\nabla_{\mathbf{x}}\mathbf{z}_j^i)^\top\bar{\mathbf{x}} + (\mathbf{z}_j^i - (\nabla_{\mathbf{x}}\mathbf{z}_j^i)^\top\mathbf{x}) = 0\}$.*

To compute the $\ell_2$ distance between $\mathbf{x}$ and the hyperplane induced by a neuron $\mathbf{z}_j^i$, we evaluate $|(\nabla_{\mathbf{x}}\mathbf{z}_j^i)^\top\mathbf{x} + (\mathbf{z}_j^i - (\nabla_{\mathbf{x}}\mathbf{z}_j^i)^\top\mathbf{x})|/\|\nabla_{\mathbf{x}}\mathbf{z}_j^i\|_2 = |\mathbf{z}_j^i|/\|\nabla_{\mathbf{x}}\mathbf{z}_j^i\|_2$. If we denote $\mathcal{I}$ as the set of hidden neuron indices $\{(i,j) | i \in [M], j \in [N_i]\}$, then $\hat{\epsilon}_{\mathbf{x},2}$ can be computed as $\hat{\epsilon}_{\mathbf{x},2} = \min_{(i,j) \in \mathcal{I}} |\mathbf{z}_j^i|/\|\nabla_{\mathbf{x}}\mathbf{z}_j^i\|_2$,

---

[2]Similar characterization also appeared in (Balestriero & Baraniuk, 2018) and, concurrently with our work, (Croce et al., 2019).

[3]When $\mathbf{x}$ has multiple possible activation patterns $\mathcal{O}$, $\hat{\epsilon}_{\mathbf{x},p}$ is always 0 regardless of the choice of $\mathcal{O}$.

where all the $\mathbf{z}_j^i$ can be computed by a single forward pass.[4] We will show in §4.1 that all the $\nabla_{\mathbf{x}}\mathbf{z}_j^i$ can also be computed efficiently by forward passes in parallel. We refer readers to Figure 1c to see a visualization of the certificates on $\ell_2$ margins.

### 3.2.3 The Number of Complete Linear Regions

The sizes of linear regions are related to their overall number, especially if we consider a bounded input space. Counting the number of linear regions in $f_\theta$ is, however, intractable due to the combinatorial nature of the activation patterns (Serra et al., 2018). We argue that counting the number of linear regions on the whole space does not capture the structure of data manifold, and we propose to certify the number of complete linear regions (#CLR) of $f_\theta$ among the data points $\mathcal{D}_{\mathbf{x}}$, which turns out to be efficient to compute given a mild condition. Here we use $\#\mathcal{A}$ to denote the cardinality of a set $\mathcal{A}$, and we have

**Lemma 7.** *(Complete Linear Region Certificate) If every data point $\mathbf{x} \in \mathcal{D}_{\mathbf{x}}$ has only one feasible activation pattern denoted as $\mathcal{O}(\mathbf{x})$, the number of complete linear regions of $f_\theta$ among $\mathcal{D}_{\mathbf{x}}$ is upperbounded by the number of different activation patterns $\#\{\mathcal{O}(\mathbf{x})|\mathbf{x} \in \mathcal{D}_{\mathbf{x}}\}$, and lower-bounded by the number of different Jacobians $\#\{J_{\mathbf{x}}f_\theta(\mathbf{x})|\mathbf{x} \in \mathcal{D}_{\mathbf{x}}\}$.*

### 3.3 Learning: Maximizing the Margins of Stable Derivatives

In this section, we focus on methods aimed at maximizing the $\ell_2$ margin $\hat{\epsilon}_{\mathbf{x},2}$, since it is (sub-)differentiable. We first formulate a regularization problem in the objective to maximize the margin:

$$\min_\theta \sum_{(\mathbf{x},\mathbf{y})\in\mathcal{D}} \left[ \mathcal{L}(f_\theta(\mathbf{x}),\mathbf{y}) - \lambda \min_{(i,j)\in\mathcal{I}} \frac{|\mathbf{z}_j^i|}{\|\nabla_{\mathbf{x}}\mathbf{z}_j^i\|_2} \right] \tag{4}$$

However, the objective itself is rather rigid due to the inner-minimization and the reciprocal of $\|\nabla_{\mathbf{x}}\mathbf{z}_j^i\|_2$. Qualitatively, such rigid loss surface hinders optimization and may attend infinity. To alleviate the problem, we do a hinge-based relaxation to the distance function similar to SVM.

### 3.3.1 Relaxation

An ideal relaxation of Eq. (4) is to disentangle $|\mathbf{z}_j^i|$ and $\|\nabla_{\mathbf{x}}\mathbf{z}_j^i\|_2$ for a smoother problem. Our first attempt is to formulate an equivalent problem with special constraints which we can leverage.

**Lemma 8.** *If there exists a (global) optimal solution of Eq. (4) that satisfies $\min_{(i,j)\in\mathcal{I}}|\mathbf{z}_j^i| > 0, \forall(\mathbf{x},\mathbf{y}) \in \mathcal{D}$, then every optimal solution of Eq. (5) is also optimal for Eq. (4).*

$$\min_\theta \sum_{(\mathbf{x},\mathbf{y})\in\mathcal{D}} \mathcal{L}(f_\theta(\mathbf{x}),\mathbf{y}) - \lambda \min_{(i,j)\in\mathcal{I}} \frac{|\mathbf{z}_j^i|}{\|\nabla_{\mathbf{x}}\mathbf{z}_j^i\|_2}, \quad s.t. \min_{(i,j)\in\mathcal{I}}|\mathbf{z}_j^i| \geq 1, \forall(\mathbf{x},\mathbf{y}) \in \mathcal{D}. \tag{5}$$

If the condition in Lemma 8 does not hold, Eq. (5) is still a valid upper bound of Eq. (4) due to a smaller feasible set. An upper bound of Eq. (5) can be obtained consequently due to the constraints:

$$\min_\theta \sum_{(\mathbf{x},\mathbf{y})\in\mathcal{D}} \mathcal{L}(f_\theta(\mathbf{x}),\mathbf{y}) - \lambda \min_{(i,j)\in\mathcal{I}} \frac{1}{\|\nabla_{\mathbf{x}}\mathbf{z}_j^i\|_2}, \quad s.t. \min_{(i,j)\in\mathcal{I}}|\mathbf{z}_j^i| \geq 1, \forall(\mathbf{x},\mathbf{y}) \in \mathcal{D}. \tag{6}$$

We then derive a relaxation that solves a smoother problem by relaxing the squared root and reciprocal on the $\ell_2$ norm as well as the hard constraint with a hinge loss to a soft regularization problem:

$$\min_\theta \sum_{(\mathbf{x},\mathbf{y})\in\mathcal{D}} \mathcal{L}(f_\theta(\mathbf{x}),\mathbf{y}) + \lambda \max_{(i,j)\in\mathcal{I}} \left[ \|\nabla_{\mathbf{x}}\mathbf{z}_j^i\|_2^2 + C\max(0, 1-|\mathbf{z}_j^i|) \right], \tag{7}$$

where $C$ is a hyper-parameter. The relaxed regularization problem can be regarded as a maximum aggregation of TSVM losses among all the neurons, where a TSVM loss with only unannotated data $\mathcal{D}_{\mathbf{x}}$ can be written as:

$$\min_{\mathbf{w},b} \sum_{\mathbf{x}\in\mathcal{D}_{\mathbf{x}}} \|\mathbf{w}\|_2^2 + C\max(0, 1-|\mathbf{w}^T\mathbf{x}+b|), \tag{8}$$

---

[4]Concurrently, Croce et al. (2019) find that the $\ell_p$ margin $\hat{\epsilon}_{\mathbf{x},p}$ can be similarly computed as $\min_{(i,j)\in\mathcal{I}} |\mathbf{z}_j^i|/\|\nabla_{\mathbf{x}}\mathbf{z}_j^i\|_q$, where $\|\cdot\|_q$ is the dual norm of the $\ell_p$-norm.

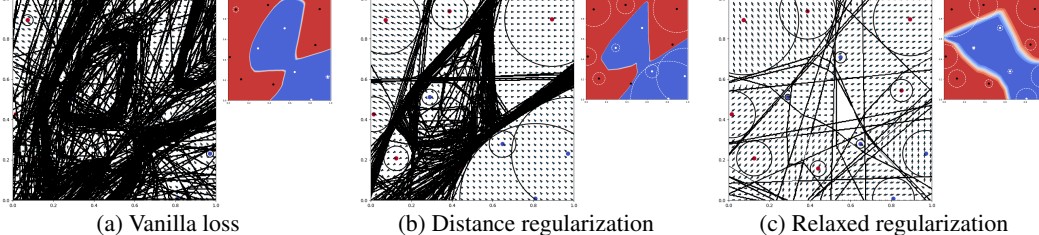

|            |            |            |
|:----------:|:----------:|:----------:|
| (a) Vanilla loss | (b) Distance regularization | (c) Relaxed regularization |

Figure 1: Toy examples of a synthetic 2D classification task. For each model (regularization type), we show a prediction heatmap (smaller pane) and the corresponding locally linear regions. The boundary of each linear region is plotted with line segments, and each circle shows the $\ell_2$ margin $\hat{\epsilon}_{\mathbf{x},2}$ around the training point. The gradient is annotated as arrows with length proportional to its $\ell_2$ norm.

which pursues a similar goal to maximize the $\ell_2$ margin in a linear model scenario, where the margin is computed between a linear hyperplane (the classifier) and the training points.

To visualize the effect of the proposed methods, we make a toy 2D binary classification dataset, and train a 4-layer fully connected network with 1) (vanilla) binary cross-entropy loss $\mathcal{L}(\cdot, \cdot)$, 2) distance regularization as in Eq. (4), and 3) relaxed regularization as in Eq. (7). Implementation details are in Appendix F. The resulting piecewise linear regions and prediction heatmaps along with gradient $\nabla_{\mathbf{x}} f_\theta(\mathbf{x})$ annotations are shown in Figure 1. The distance regularization enlarges the linear regions around each training point, and the relaxed regularization further generalizes the property to the whole space; the relaxed regularization possesses a smoother prediction boundary, and has a special central region where the gradients are 0 to allow gradients to change directions smoothly.

### 3.3.2 IMPROVING SPARSE LEARNING SIGNALS

Since a linear region is shaped by a set of neurons that are "close" to a given a point, a noticeable problem of Eq. (7) is that it only focuses on the "closest" neuron, making it hard to scale the effect to large networks. Hence, we make a generalization to the relaxed loss in Eq. (7) with a set of neurons that incur high losses to the given point. We denote $\hat{\mathcal{I}}(\mathbf{x}, \gamma)$ as the set of neurons with top $\gamma$ percent relaxed loss (TSVM loss) on $\mathbf{x}$. The generalized loss is our final objective for learning RObust Local Linearity (ROLL) and is written as:

$$\min_\theta \sum_{(\mathbf{x},\mathbf{y}) \in \mathcal{D}} \mathcal{L}(f_\theta(\mathbf{x}), \mathbf{y}) + \frac{\lambda}{\#\hat{\mathcal{I}}(\mathbf{x}, \gamma)} \sum_{(i,j) \in \hat{\mathcal{I}}(\mathbf{x}, \gamma)} \left[ \|\nabla_{\mathbf{x}} \mathbf{z}_j^i\|_2^2 + C \max(0, 1 - |\mathbf{z}_j^i|) \right]. \qquad (9)$$

A special case of Eq. (9) is when $\gamma = 100$ (i.e. $\hat{\mathcal{I}}(\mathbf{x}, 100) = \mathcal{I}$), where the nonlinear sorting step effectively disappears. Such simple additive structure without a nonlinear sorting step can stabilize the training process, is simple to parallelize computation, and allows for an approximate learning algorithm as will be developed in §4.2. Besides, taking $\gamma = 100$ can induce a strong synergy effect, as all the gradient norms $\|\nabla_{\mathbf{x}} \mathbf{z}_j^i\|_2^2$ in Eq. (9) between any two layers are highly correlated.

## 4 COMPUTATION, APPROXIMATE LEARNING, AND COMPATIBILITY

### 4.1 PARALLEL COMPUTATION OF GRADIENTS

The $\ell_2$ margin $\hat{\epsilon}_{\mathbf{x},2}$ and the ROLL loss in Eq. (9) demands heavy computation on gradient norms. While calling back-propagation $|\mathcal{I}|$ times is intractable, we develop a parallel algorithm without calling a single back-propagation by exploiting the functional structure of $f_\theta$.

Given an activation pattern, we know that each hidden neuron $\mathbf{z}_j^i$ is also a linear function of $\mathbf{x} \in S(\mathbf{x})$. We can construct another *linear* network $g_\theta$ that is identical to $f_\theta$ in $S(\mathbf{x})$ based on the same set of parameters but fixed linear activation functions constructed to mimic the behavior of $f_\theta$ in $S(\mathbf{x})$. Due to the linearity of $g_\theta$, the derivatives of all the neurons to an input axis can be computed by forwarding two samples: subtracting the neurons with an one-hot input from the same neurons with a zero input. The procedure can be amortized and parallelized to all the dimensions by feeding

Table 1: FC networks on MNIST dataset. #CLR is the number of complete linear regions among the 10K testing points, and $\hat{\epsilon}_{\mathbf{x},p}$ shows the $\ell_p$ margin for each $r \in \{25, 50, 75, 100\}$ percentile $P_r$.

| Loss | $C$ | ACC | #CLR | $\hat{\epsilon}_{\mathbf{x},1}(\times 10^{-4})$ | | | | $\hat{\epsilon}_{\mathbf{x},2}(\times 10^{-4})$ | | | |
|------|-----|-----|------|------|------|------|------|------|------|------|------|
| | | | | $P_{25}$ | $P_{50}$ | $P_{75}$ | $P_{100}$ | $P_{25}$ | $P_{50}$ | $P_{75}$ | $P_{100}$ |
| Vanilla | | 98% | 10000 | 22 | 53 | 106 | 866 | 3 | 6 | 13 | 91 |
| ROLL | 0.25 | 98% | 9986 | 219 | 530 | 1056 | 6347 | 37 | 92 | 182 | 1070 |
| ROLL | 1.00 | 97% | 8523 | 665 | 1593 | 3175 | 21825 | 125 | 297 | 604 | 4345 |

$D + 1$ samples to $g_\theta$ in parallel. We remark that the algorithm generalizes to all the piecewise linear networks, and refer readers to Appendix C for algorithmic details.[5]

To analyze the complexity of the proposed approach, we assume that parallel computation does not incur any overhead and a batch matrix multiplication takes a unit operation. To compute the gradients of all the neurons for a batch of inputs, our perturbation algorithm takes $2M$ operations, while back-propagation takes $\sum_{i=1}^{M} 2iN_i$ operations. The detailed analysis is also in Appendix C.

### 4.2 APPROXIMATE LEARNING

Despite the parallelizable computation of $\nabla_{\mathbf{x}}\mathbf{z}_j^i$, it is still challenging to compute the loss for large networks in a high dimension setting, where even calling $D + 1$ forward passes in parallel as used in §4.1 is infeasible due to memory constraints. Hence we propose an unbiased estimator of the ROLL loss in Eq. (9) when $\hat{\mathcal{I}}(\mathbf{x}, \gamma) = \mathcal{I}$. Note that $\sum_{(i,j)\in\mathcal{I}} C \max(0, 1 - |\mathbf{z}_j^i|)$ is already computable in one single forward pass. For the sum of gradient norms, we use the following equivalent decoupling:

$$\frac{1}{\#\mathcal{I}} \sum_{(i,j)\in\mathcal{I}} \|\nabla_{\mathbf{x}}\mathbf{z}_j^i\|_2^2 = \frac{1}{\#\mathcal{I}} \sum_{k=1}^{D} \sum_{(i,j)\in\mathcal{I}} (\frac{\partial \mathbf{z}_j^i}{\partial \mathbf{x}_k})^2 = \frac{D}{\#\mathcal{I}} \mathbb{E}_{k\sim\text{Unif}([D])} \left[ \sum_{(i,j)\in\mathcal{I}} (\frac{\partial \mathbf{z}_j^i}{\partial \mathbf{x}_k})^2 \right], \qquad (10)$$

where the summation inside the expectation in the last equation can be efficiently computed using the procedure in §4.1 and is in general storable within GPU memory. In practice, we can uniformly sample $D'$ ($1 \leq D' \ll D$) input axes to have an unbiased approximation to Eq. (10), where computing all the partial derivatives with respect to $D'$ axes only requires $D' + 1$ times memory (one hot vectors and a zero vector) than a typical forward pass for $\mathbf{x}$.

### 4.3 COMPATIBILITY

The proposed algorithms can be used on all the deep learning models with affine transformations and piecewise linear activation functions by enumerating every neuron that will be imposed an ReLU-like activation function as $\mathbf{z}_j^i$. They do not immediately generalize to the nonlinearity of maxout/max-pooling (Goodfellow et al., 2013) that also yields a piecewise linear function. We provide an initial step towards doing so in the Appendix E, but we suggest to use an average-pooling or convolution with large strides instead, since they do not induce extra linear constraints as max-pooling and do not in general yield significant difference in performance (Springenberg et al., 2014).

## 5 EXPERIMENTS

In this section, we compare our approach ('ROLL') with a baseline model with the same training procedure except the regularization ('vanilla') in several scenarios. All the reported quantities are computed on a testing set. Experiments are run on single GPU with 12G memory.

### 5.1 MNIST

**Evaluation Measures:** 1) accuracy (ACC), 2) number of complete linear regions (#CLR), and 3) $\ell_p$ margins of linear regions $\hat{\epsilon}_{\mathbf{x},p}$. We compute the margin $\hat{\epsilon}_{\mathbf{x},p}$ for each testing point $\mathbf{x}$ with $p \in \{1, 2\}$, and we evaluate $\hat{\epsilon}_{\mathbf{x},p}$ on 4 different percentiles $P_{25}, P_{50}, P_{75}, P_{100}$ among the testing data.

---

[5]When the network is FC (or can efficiently be represented as such), one can use a dynamic programming algorithm to compute the gradients (Papernot et al., 2016), which is included in Appendix D.

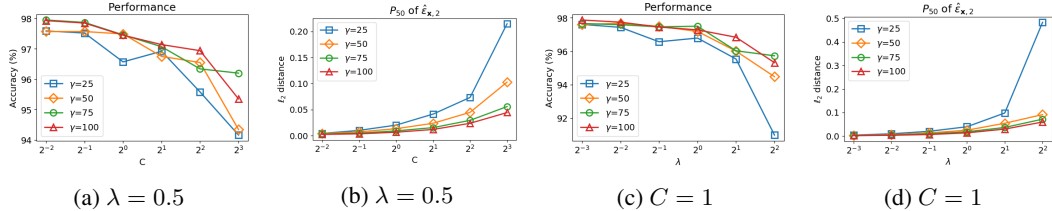

|        | (a) $\lambda = 0.5$ | (b) $\lambda = 0.5$ | (c) $C = 1$ | (d) $C = 1$ |
|--------|---------------------|---------------------|-------------|-------------|

Figure 2: Parameter analysis on MNIST dataset. $P_{50}$ of $\hat{\epsilon}_{\mathbf{x},2}$ is the median of $\hat{\epsilon}_{\mathbf{x},2}$ in the testing data.

Table 2: Running time for a gradient descent step of FC networks on MNIST dataset. The full setting refers to Eq. (9) ($\gamma = 100$), and 3-samples refers to approximating Eq. (10) with 3 samples.

|                           | Vanilla | ROLL (full; back-prop) | ROLL (full; perturb) | ROLL (3-samples; perturb) |
|---------------------------|---------|------------------------|----------------------|---------------------------|
| second ($\times 10^{-5}$) | 129     | 31185                  | 2667                 | 298                       |

We use a $55,000/5,000/10,000$ split of MNIST dataset for training/validation/testing. Experiments are conducted on a 4-layer FC model with ReLU activations. The implementation details are in Appendix G. We report the two models with the largest median $\hat{\epsilon}_{\mathbf{x},2}$ among validation data given the same and $1\%$ less validation accuracy compared to the baseline model.

The results are shown in Table 1. The tuned models have $\gamma = 100, \lambda = 2$, and different $C$ as shown in the table. The condition in Lemma 7 for certifying #CLR is satisfied with tight upper bound and lower bound, so a single number is reported. Given the same performance, the ROLL loss achieves about 10 times larger margins for most of the percentiles than the vanilla loss. By trading-off $1\%$ accuracy, about 30 times larger margins can be achieved. The Spearman's rank correlation between $\hat{\epsilon}_{\mathbf{x},1}$ and $\hat{\epsilon}_{\mathbf{x},2}$ among testing data is at least 0.98 for all the cases. The lower #CLR in our approach than the baseline model reflects the existence of certain larger linear regions that span across different testing points. All the points inside the same linear region in the ROLL model with ACC$= 98\%$ have the same label, while there are visually similar digits (e.g., 1 and 7) in the same linear region in the other ROLL model. We do a parameter analysis in Figure 2 with the ACC and $P_{50}$ of $\hat{\epsilon}_{\mathbf{x},2}$ under different $C, \lambda$ and $\gamma$ when the other hyper-parameters are fixed. As expected, with increased $C$ and $\lambda$, the accuracy decreases with an increased $\ell_2$ margin. Due to the smoothness of the curves, higher $\gamma$ values reflect less sensitivity to hyper-parameters $C$ and $\lambda$.

To validate the efficiency of the proposed method, we measure the running time for performing a complete mini-batch gradient descent step (starting from the forward pass) on average. We compare 1) the vanilla loss, 2) the full ROLL loss ($\gamma = 100$) in Eq. (9) computed by back-propagation, 3) the same as 2) but computed by our perturbation algorithm, and 4) the approximate ROLL loss in Eq. (10) computed by perturbations. The approximation is computed with $3 = D/256$ samples. The results are shown in Table 2. The accuracy and $\ell_2$ margins of the approximate ROLL loss are comparable to the full loss. Overall, our approach is only twice slower than the vanilla loss. The approximate loss is about 9 times faster than the full loss. Compared to back-propagation, our perturbation algorithm achieves about 12 times empirical speed-up. In summary, the computational overhead of our method is minimal compared to the vanilla loss, which is achieved by the perturbation algorithm and the approximate loss.

Table 3: RNNs on the Japanese Vowel dataset. $\hat{\epsilon}_{\mathbf{x},p}$ shows the $\ell_p$ margin for each $r \in \{25, 50, 75, 100\}$ percentile $P_r$ in the testing data (the larger the better).

| Loss    | $\lambda$ | $C$   | ACC   | $\hat{\epsilon}_{\mathbf{x},1}(\times 10^{-6})$ | | | | $\hat{\epsilon}_{\mathbf{x},2}(\times 10^{-6})$ | | | |
|---------|-----------|-------|-------|----------|----------|----------|-----------|----------|----------|----------|-----------|
|         |           |       |       | $P_{25}$ | $P_{50}$ | $P_{75}$ | $P_{100}$ | $P_{25}$ | $P_{50}$ | $P_{75}$ | $P_{100}$ |
| Vanilla |           |       | $98\%$ | 66       | 177      | 337      | 1322      | 23       | 61       | 113      | 438       |
| ROLL    | $2^{-5}$  | $2^4$ | $98\%$ | 264      | 562      | 1107     | 5227      | 95       | 207      | 407      | 1809      |
| ROLL    | $2^{-1}$  | $2^2$ | $97\%$ | 1284     | 2898     | 6086     | 71235     | 544      | 1249     | 2644     | 22968     |

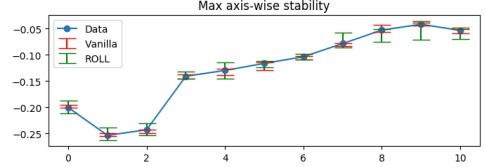 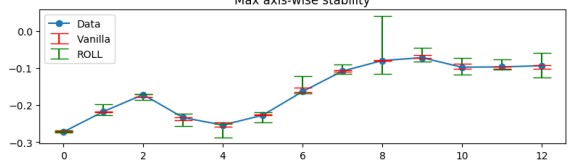

(a) The 9th channel of the sequence that yields $P_{50}$ of $\hat{\epsilon}_{\mathbf{x},2}$ on the ROLL model.

(b) The 9th channel of the sequence that yields $P_{75}$ of $\hat{\epsilon}_{\mathbf{x},2}$ on the ROLL model.

Figure 3: Stability bounds on derivatives on the Japanese Vowel dataset.

Table 4: ResNet on Caltech-256. Here $\Delta(\mathbf{x}, \mathbf{x}', \mathbf{y})$ denotes $\ell_1$ gradient distortion $\|\nabla_{\mathbf{x}'} f_\theta(\mathbf{x}')_{\mathbf{y}} - \nabla_{\mathbf{x}} f_\theta(\mathbf{x})_{\mathbf{y}}\|_1$ (the smaller the better for each $r$ percentile $P_r$ among the testing data).

| Loss | P@1 | P@5 | $\mathbb{E}_{\mathbf{x}' \sim \text{Unif}(\bar{\mathcal{B}}_{\epsilon,\infty}(\mathbf{x}))}[\Delta(\mathbf{x}, \mathbf{x}', \mathbf{y})]$ | | | | $\max_{\mathbf{x}' \in \bar{\mathcal{B}}_{\epsilon,\infty}(\mathbf{x})}[\Delta(\mathbf{x}, \mathbf{x}', \mathbf{y})]$ | | | |
|---|---|---|---|---|---|---|---|---|---|---|
| | | | $P_{25}$ | $P_{50}$ | $P_{75}$ | $P_{100}$ | $P_{25}$ | $P_{50}$ | $P_{75}$ | $P_{100}$ |
| Vanilla | 80.7% | 93.4% | 583.8 | 777.4 | 1041.9 | 3666.7 | 840.9 | 1118.2 | 1477.6 | 5473.5 |
| ROLL | 80.8% | 94.1% | 540.6 | 732.0 | 948.7 | 2652.2 | 779.9 | 1046.7 | 1368.2 | 3882.8 |

## 5.2 SPEAKER IDENTIFICATION

We train RNNs for speaker identification on a Japanese Vowel dataset from the UCI machine learning repository (Dheeru & Karra Taniskidou, 2017) with the official training/testing split.[6] The dataset has variable sequence length between 7 and 29 with 12 channels and 9 classes. We implement the network with the state-of-the-art scaled Cayley orthogonal RNN (scoRNN) (Helfrich et al., 2018), which parameterizes the transition matrix in RNN using orthogonal matrices to prevent gradient vanishing/exploding, with LeakyReLU activation. The implementation details are in Appendix H. The reported models are based on the same criterion as §5.1.

The results are reported in Table 3. With the same/1% inferior ACC, our approach leads to a model with about 4/20 times larger margins among the percentiles on testing data, compared to the vanilla loss. The Spearman's rank correlation between $\hat{\epsilon}_{\mathbf{x},1}$ and $\hat{\epsilon}_{\mathbf{x},2}$ among all the cases are 0.98. We also conduct sensitivity analysis on the derivatives by finding $\hat{\epsilon}_{\mathbf{x},\Delta\mathbf{x}}$ along each coordinate $\Delta\mathbf{x} \in \cup_i \cup_{j=1}^{12} \{-\mathbf{e}^{i,j}, \mathbf{e}^{i,j}\}$ ($\mathbf{e}_{k,l}^{i,j} = 0, \forall k, l$ except $\mathbf{e}_{i,j}^{i,j} = 1$), which identifies the stability bounds $[\hat{\epsilon}_{\mathbf{x},-\mathbf{e}^{i,j}}, \hat{\epsilon}_{\mathbf{x},\mathbf{e}^{i,j}}]$ at each timestamp $i$ and channel $j$ that guarantees stable derivatives. The visualization using the vanilla and our ROLL model with 98% ACC is in Figure 3. Qualitatively, the stability bound of the ROLL regularization is consistently larger than the vanilla model.

## 5.3 CALTECH-256

We conduct experiments on Caltech-256 (Griffin et al., 2007), which has 256 classes, each with at least 80 images. We downsize the images to $299 \times 299 \times 3$ and train a 18-layer ResNet (He et al., 2016) with initializing from parameters pre-trained on ImageNet (Deng et al., 2009). The approximate ROLL loss in Eq. (10) is used with 120 random samples on each channel. We randomly select 5 and 15 samples in each class as the validation and testing set, respectively, and put the remaining data into the training set. The implementation details are in Appendix I.

**Evaluation Measures:** Due to high input dimensionality ($D \approx 270K$), computing the certificates $\hat{\epsilon}_{\mathbf{x},1}, \hat{\epsilon}_{\mathbf{x},2}$ is computationally challenging without a cluster of GPUs. Hence, we turn to a sample-based approach to evaluate the stability of the gradients $f_\theta(\mathbf{x})_{\mathbf{y}}$ for the ground-truth label in a local region with a goal to reveal the stability across different linear regions. Note that evaluating the gradient of the prediction instead is problematic to compare different models in this case.

Given labeled data $(\mathbf{x}, \mathbf{y})$, we evaluate the stability of gradient $\nabla_{\mathbf{x}} f_\theta(\mathbf{x})_{\mathbf{y}}$ in terms of expected $\ell_1$ distortion (over a uniform distribution) and the maximum $\ell_1$ distortion within the intersection $\bar{\mathcal{B}}_{\epsilon,\infty}(\mathbf{x}) = \mathcal{B}_{\epsilon,\infty}(\mathbf{x}) \cap \mathcal{X}$ of an $\ell_\infty$-ball and the domain of images $\mathcal{X} = [0, 1]^{299 \times 299 \times 3}$. The $\ell_1$ gradient distortion is defined as $\Delta(\mathbf{x}, \mathbf{x}', \mathbf{y}) := \|\nabla_{\mathbf{x}'} f_\theta(\mathbf{x}')_{\mathbf{y}} - \nabla_{\mathbf{x}} f_\theta(\mathbf{x})_{\mathbf{y}}\|_1$. For a fixed $\mathbf{x}$, we refer to

---

[6]The parameter is tuned on the testing set and thus the performance should be interpreted as validation.

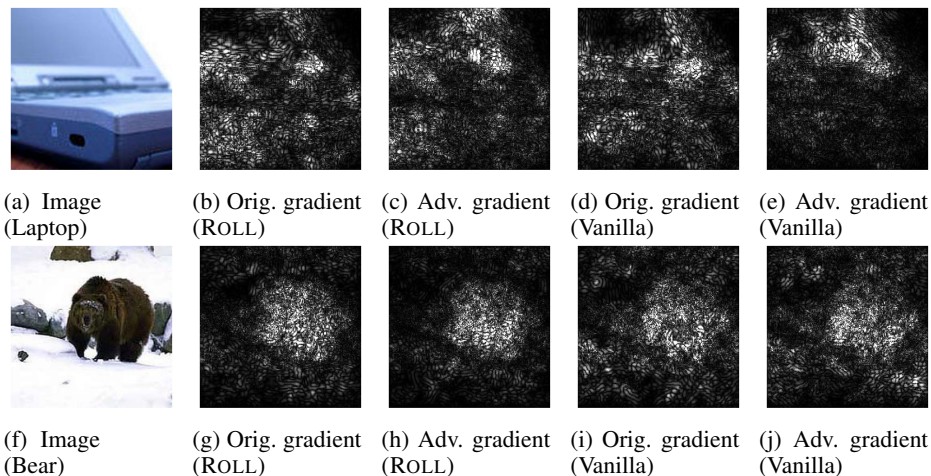

| (a) Image (Laptop) | (b) Orig. gradient (ROLL) | (c) Adv. gradient (ROLL) | (d) Orig. gradient (Vanilla) | (e) Adv. gradient (Vanilla) |
| (f) Image (Bear) | (g) Orig. gradient (ROLL) | (h) Adv. gradient (ROLL) | (i) Orig. gradient (Vanilla) | (j) Adv. gradient (Vanilla) |

Figure 4: Visualization of the examples in Caltech-256 that yield the $P_{50}$ (above) and $P_{75}$ (below) of the maximum $\ell_1$ gradient distortions among the testing data on our ROLL model. The adversarial gradient is found by maximizing the distortion $\Delta(\mathbf{x}, \mathbf{x}', \mathbf{y})$ over the $\ell_\infty$-norm ball with radius $8/256$.

the maximizer $\nabla_{\mathbf{x}'} f_\theta(\mathbf{x}')_{\mathbf{y}}$ as the *adversarial gradient*. Computation of the maximum $\ell_1$ distortion requires optimization, but gradient-based optimization is not applicable since the gradient of the loss involves the Hessian $\nabla^2_{\mathbf{x}'} f_\theta(\mathbf{x}')_{\mathbf{y}}$ which is either $0$ or ill-defined due to piecewise linearity. Hence, we use a genetic algorithm (Whitley, 1994) for black-box optimization. Implementation details are provided in Appendix J. We use $8000$ samples to approximate the expected $\ell_1$ distortion. Due to computational limits, we only evaluate $1024$ random images in the testing set for both maximum and expected $\ell_1$ gradient distortions. The $\ell_\infty$-ball radius $\epsilon$ is set to $8/256$.

The results along with precision at 1 and 5 (P@1 and P@5) are presented in Table 4. The ROLL loss yields more stable gradients than the vanilla loss with marginally superior precisions. Out of $1024$ examined examples $\mathbf{x}$, only $40$ and $42$ gradient-distorted images change prediction labels in the ROLL and vanilla model, respectively. We visualize some examples in Figure 4 with the original and adversarial gradients for each loss. Qualitatively, the ROLL loss yields stable shapes and intensities of gradients, while the vanilla loss does not. More examples with integrated gradient attributions (Sundararajan et al., 2017) are provided in Appendix K.

## 6 CONCLUSION

This paper introduces a new learning problem to endow deep learning models with robust local linearity. The central attempt is to construct locally transparent neural networks, where the derivatives faithfully approximate the underlying function and lends itself to be stable tools for further applications. We focus on piecewise linear networks and solve the problem based on a margin principle similar to SVM. Empirically, the proposed ROLL loss expands regions with provably stable derivatives, and further generalize the stable gradient property across linear regions.

### ACKNOWLEDGMENTS

The authors acknowledge support for this work by a grant from Siemens Corporation, thank the anonymous reviewers for their helpful comments, and thank Hao He and Yonglong Tian for helpful discussions.

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

# A   PROOFS

## A.1   PROOF OF LEMMA 2

**Lemma 2.** *Given an activation pattern $\mathcal{O}$ with any feasible point $\mathbf{x}$, each activation indicator $\mathbf{o}_j^i \in \mathcal{O}$ induces a feasible set $S_j^i(\mathbf{x}) = \{\bar{\mathbf{x}} \in \mathbb{R}^D : \mathbf{o}_j^i[(\nabla_{\mathbf{x}} \mathbf{z}_j^i)^\top \bar{\mathbf{x}} + (\mathbf{z}_j^i - (\nabla_{\mathbf{x}} \mathbf{z}_j^i)^\top \mathbf{x})] \geq 0\}$, and the feasible set of the activation pattern is equivalent to $S(\mathbf{x}) = \cap_{i=1}^M \cap_{j=1}^{N_i} S_j^i(\mathbf{x})$.*

*Proof.* For $j \in [N_1]$, we have $\nabla_{\mathbf{x}} \mathbf{z}_j^1 = \mathbf{W}_{j,:}^1$, $(\mathbf{z}_j^1 - (\nabla_{\mathbf{x}} \mathbf{z}_j^1)^\top \mathbf{x}) = \mathbf{b}_j^1$. If $\bar{\mathbf{x}}$ is feasible to the fixed activation pattern $\mathbf{o}_j^1$, it is equivalent to that $\bar{\mathbf{x}}$ satisfies the linear constraint

$$\mathbf{o}_j^1[(\nabla_{\mathbf{x}} \mathbf{z}_j^i)^\top \bar{\mathbf{x}} + (\mathbf{z}_j^1 - (\nabla_{\mathbf{x}} \mathbf{z}_j^1)^\top \mathbf{x})] = \mathbf{o}_j^1[(\mathbf{W}_{j,:}^1)^\top \bar{\mathbf{x}} + \mathbf{b}_j^1] \geq 0 \tag{11}$$

in the first layer.

Assume $\bar{\mathbf{x}}$ has satisfied all the constraints before layer $i > 1$. We know if all the previous layers follows the fixed activation indicators, it is equivalent to rewrite each

$$\mathbf{a}_{j'}^{i'} = \max(0, \mathbf{o}_{j'}^{i'}) \cdot \mathbf{z}_{j'}^{i'}, \forall j' \in [N_{i'}], i' \in [i-1]. \tag{12}$$

Then for $j \in [N_i]$, it is clear that $\mathbf{z}_j^i$ is a fixed linear function of $\mathbf{x}$ with linear weights equal to $\nabla_{\mathbf{x}} \mathbf{z}_j^i$ by construction. If $\bar{\mathbf{x}}$ is also feasible to the fixed activation indicator $\mathbf{o}_j^i$, it is equivalent to that $\bar{\mathbf{x}}$ also satisfies the linear constraint

$$\mathbf{o}_j^i[(\nabla_{\mathbf{x}} \mathbf{z}_j^i)^\top \bar{\mathbf{x}} + (\mathbf{z}_j^i - (\nabla_{\mathbf{x}} \mathbf{z}_j^i)^\top \mathbf{x})] \geq 0. \tag{13}$$

The proof follows by induction. $\square$

## A.2   PROOF OF PROPOSITION 4

**Proposition 4.** *(Directional Feasibility) Given a point $\mathbf{x}$, a feasible set $S(\mathbf{x})$ and a unit vector $\Delta\mathbf{x}$, if $\exists \bar{\epsilon} \geq 0$ such that $\mathbf{x} + \bar{\epsilon}\Delta\mathbf{x} \in S(\mathbf{x})$, then $f_\theta$ is linear in $\{\mathbf{x} + \epsilon\Delta\mathbf{x} : 0 \leq \epsilon \leq \bar{\epsilon}\}$.*

*Proof.* Since $S(\mathbf{x})$ is a convex set and $\mathbf{x}, \mathbf{x} + \bar{\epsilon}\Delta\mathbf{x} \in S(\mathbf{x})$, $\{\mathbf{x} + \epsilon\Delta\mathbf{x} : 0 \leq \epsilon \leq \bar{\epsilon}\} \subseteq S(\mathbf{x})$. $\square$

## A.3   PROOF OF PROPOSITION 5

**Proposition 5.** *($\ell_1$-ball Feasibility) Given a point $\mathbf{x}$, a feasible set $S(\mathbf{x})$, and an $\ell_1$-ball $\mathcal{B}_{\epsilon,1}(\mathbf{x})$ with extreme points $\mathbf{x}^1, \ldots, \mathbf{x}^{2D}$, if $\mathbf{x}^i \in S(\mathbf{x}), \forall i \in [2D]$, then $f_\theta$ is linear in $\mathcal{B}_{\epsilon,1}(\mathbf{x})$.*

*Proof.* $S(\mathbf{x})$ is a convex set and $\mathbf{x}^i \in S(\mathbf{x}), \forall i \in [2D]$. Hence, $\forall \mathbf{x}' \in \mathcal{B}_{\epsilon,1}(\mathbf{x})$, we know $\mathbf{x}'$ is a convex combination of $\mathbf{x}^1, \ldots, \mathbf{x}^{2D}$, which implies $\mathbf{x}' \in S(\mathbf{x})$. $\square$

## A.4   PROOF OF PROPOSITION 6

**Proposition 6.** *($\ell_2$-ball Certificate) Given a point $\mathbf{x}$, $\hat{\epsilon}_{\mathbf{x},2}$ is the minimum $\ell_2$ distance between $\mathbf{x}$ and the union of hyperplanes $\cup_{i=1}^M \cup_{j=1}^{N_i} \{\bar{\mathbf{x}} \in \mathbb{R}^D : (\nabla_{\mathbf{x}} \mathbf{z}_j^i)^\top \bar{\mathbf{x}} + (\mathbf{z}_j^i - (\nabla_{\mathbf{x}} \mathbf{z}_j^i)^\top \mathbf{x}) = 0\}$.*

*Proof.* Since $S(\mathbf{x})$ is a convex polyhedron and $\mathbf{x} \in S(\mathbf{x})$, $\mathcal{B}_{\epsilon,2}(\mathbf{x}) \subseteq S(\mathbf{x})$ is equivalent to the statement: the hyperplanes induced from the linear constraints in $S(\mathbf{x})$ are away from $\mathbf{x}$ for at least $\epsilon$ in $\ell_2$ distance. Accordingly, the minimizing $\ell_2$ distance between $\mathbf{x}$ and the hyperplanes is the maximizing distance that satisfies $\mathcal{B}_{\epsilon,2}(\mathbf{x}) \subseteq S(\mathbf{x})$. $\square$

### A.5 PROOF OF LEMMA 7

**Lemma 7.** *(Complete Linear Region Certificate) If every data point $\mathbf{x} \in \mathcal{D}_{\mathbf{x}}$ has only one feasible activation pattern denoted as $\mathcal{O}(\mathbf{x})$, the number of complete linear regions of $f_\theta$ among $\mathcal{D}_{\mathbf{x}}$ is upper-bounded by the number of different activation patterns $\#\{\mathcal{O}(\mathbf{x}) | \mathbf{x} \in \mathcal{D}_{\mathbf{x}}\}$, and lower-bounded by the number of different Jacobians $\#\{J_{\mathbf{x}} f_\theta(\mathbf{x}) | \mathbf{x} \in \mathcal{D}_{\mathbf{x}}\}$.*

*Proof.* The number of different activation patterns is an upper bound since it counts the number of linear regions instead of the number of complete linear regions (a complete linear region can contain multiple linear regions). The number of different Jacobians is a lower bound since it only count the number of different linear coefficients $f_\theta(\mathbf{x})$ on $\mathcal{D}_{\mathbf{x}}$ without distinguishing whether they are in the same connected region. $\square$

### A.6 PROOF OF LEMMA 8

**Lemma 8.** *If there exists a (global) optimal solution of Eq. (4) that satisfies $\min_{(i,j)\in\mathcal{I}} |\mathbf{z}_j^i| > 0, \forall(\mathbf{x}, \mathbf{y}) \in \mathcal{D}$, then every optimal solution of Eq. (5) is also optimal for Eq. (4).*

$$\min_\theta \sum_{(\mathbf{x},\mathbf{y})\in\mathcal{D}} \mathcal{L}(f_\theta(\mathbf{x}), \mathbf{y}) - \lambda \min_{(i,j)\in\mathcal{I}} \frac{|\mathbf{z}_j^i|}{\|\nabla_{\mathbf{x}}\mathbf{z}_j^i\|_2}, \quad s.t. \min_{(i,j)\in\mathcal{I}} |\mathbf{z}_j^i| \geq 1, \forall(\mathbf{x}, \mathbf{y}) \in \mathcal{D}.$$

*Proof.* The proof is based on constructing a neural network feasible in Eq. (5) that has the same loss as the optimal model in Eq. (4). Since the optimum in Eq. (5) is lower-bounded by the optimum in Eq. (4) due to smaller feasible set, a model feasible in Eq. (5) and having the same loss as the optimum in Eq. (4) implies that it is also optimal in Eq. (5).

Given the optimal model $f_\theta$ in Eq. (4) satisfying the constraint $\min_{(i,j)\in\mathcal{I}} |\mathbf{z}_j^i| > 0, \forall(\mathbf{x}, \mathbf{y}) \in \mathcal{D}$, we construct a model feasible in Eq. (5). For $i = 1, ..., M$, we compute the smallest neuron response $\delta_j^i = \min_{(\mathbf{x},\mathbf{y})\in\mathcal{D}} |\mathbf{z}_j^i|$ in $f_\theta$, and revise its weights by the following rules:

$$\mathbf{W}_{j,:}^i \leftarrow \frac{1}{\delta_j^i}\mathbf{W}_{j,:}^i, \quad \mathbf{b}_j^i \leftarrow \frac{1}{\delta_j^i}\mathbf{b}_j^i, \quad \mathbf{W}_{:,j}^{i+1} \leftarrow \delta_j^i\mathbf{W}_{:,j}^{i+1}. \tag{14}$$

The above rule only scale the neuron value of $\mathbf{z}_j^i$ without changing the value of all the higher layers, so the realized function of $f_\theta$ does not change. That says, it achieves the same objective value as Eq. (4) while being feasible in Eq. (5), and thus being an optimum in both Eq. (4) and Eq. (5).

Since the other optimal solutions in Eq. (5) have the same loss as the constructed network, they are also optimal in Eq. (4). $\square$

## B CERTIFICATE OF DIRECTIONAL MARGIN AND $\ell_1$ MARGIN

To certify the directional margin $\hat{\epsilon}_{\mathbf{x},\Delta\mathbf{x}} := \max_{\{\epsilon\geq 0: \mathbf{x}+\epsilon\Delta\mathbf{x}\in S(\mathbf{x})\}} \epsilon$ along a given $\Delta\mathbf{x}$, we can do a binary search between a lower bound $\epsilon^l$ and upper bound $\epsilon^u$. We initialize $\epsilon_0^l = 0$ and run a subroutine to exponentially find an arbitrary upper bound $\epsilon_0^u$ such that $\mathbf{x} + \epsilon_0^u\Delta\mathbf{x} \notin S(\mathbf{x})$. If $\epsilon_0^u$ does not exist, we have $\hat{\epsilon}_{\mathbf{x},\Delta\mathbf{x}} = \infty$; otherwise, a binary search is executed by iterating $t$ as

$$\begin{cases} \epsilon_{t+1}^l := 0.5(\epsilon_t^l + \epsilon_t^u), \epsilon_{t+1}^u := \epsilon_t^u, & \text{if } \mathbf{x} + 0.5(\epsilon_t^l + \epsilon_t^u)\Delta\mathbf{x} \in S(\mathbf{x}) \\ \epsilon_{t+1}^l(\mathbf{x}) := \epsilon_t^l, \epsilon_{t+1}^u := 0.5(\epsilon_t^l + \epsilon_t^u), & \text{otherwise} \end{cases} \tag{15}$$

Clearly, $\mathbf{x} + \epsilon_t^l\Delta\mathbf{x} \in S(\mathbf{x}), \mathbf{x} + \epsilon_t^u\Delta\mathbf{x} \notin S(\mathbf{x})$ always holds, and the gap between $\epsilon_t^l$ and $\epsilon_t^u$ decreases exponentially fast: $\epsilon_t^u - \epsilon_t^l = 0.5^t(\epsilon_0^u - \epsilon_0^l)$, which upper-bounds $\hat{\epsilon}_{\mathbf{x},\Delta\mathbf{x}} - \epsilon_t^l \leq \epsilon_t^u - \epsilon_t^l$. In practice, we run the binary search with finite iteration $T$ and return the lower bound $\epsilon_T^l$ with an identifiable bound $\epsilon_T^u - \epsilon_T^l$ from the optimal solution. In our experiments, we run the binary search algorithm until the bound is less than $10^{-7}$.

If we denote $\mathbf{e}^1, \ldots, \mathbf{e}^D$ as the set of unit vector in each axis ($\mathbf{e}_i^i = 1, \mathbf{e}_j^i = 0, \forall j \neq i$), the margin $\hat{\epsilon}_{\mathbf{x},1}$ for the $\ell_1$-ball $\mathcal{B}_{\epsilon,1}(\mathbf{x})$ can be certified by 1) computing the set of directional margins for each extreme point direction of an $\ell_1$ ball $\{\hat{\epsilon}_{\mathbf{x},-\mathbf{e}^1}, \hat{\epsilon}_{\mathbf{x},\mathbf{e}^1}, \ldots, \hat{\epsilon}_{\mathbf{x},-\mathbf{e}^D}, \hat{\epsilon}_{\mathbf{x},\mathbf{e}^D}\}$, and 2) returning the minimum in the set as $\hat{\epsilon}_{\mathbf{x},1}$.

## C  PARALLEL COMPUTATION OF THE GRADIENTS BY LINEARITY

We denote the corresponding neurons $\mathbf{z}_j^i$ and $\mathbf{a}_j^i$ of $f_\theta$ in $g_\theta$ as $\hat{\mathbf{z}}_j^i(\hat{\mathbf{x}})$ and $\hat{\mathbf{a}}_j^i(\hat{\mathbf{x}})$ given $\hat{\mathbf{x}}$, highlighting its functional relationship with respect to a new input $\hat{\mathbf{x}}$. The network $g_\theta$ is constructed with exactly the same weights and biases as $f_\theta$ but with a well-crafted *linear* activation function $\hat{\mathbf{o}}_j^i = \max(0, \mathbf{o}_j^i) \in \{0, 1\}$. Note that since $\mathbf{o}$ is given, $\hat{\mathbf{o}}$ is fixed. Then each layer in $g_\theta$ is represented as:

$$\hat{\mathbf{a}}^i(\hat{\mathbf{x}}) = \hat{\mathbf{o}}^i \odot \hat{\mathbf{z}}^i(\hat{\mathbf{x}}), \ \ \hat{\mathbf{z}}^i(\hat{\mathbf{x}}) = \mathbf{W}^i \hat{\mathbf{a}}^{i-1}(\hat{\mathbf{x}}) + \mathbf{b}^i, \forall i \in [M]. \ \ \hat{\mathbf{a}}^0(\hat{\mathbf{x}}) = \hat{\mathbf{x}}. \tag{16}$$

We note that $\hat{\mathbf{a}}^i(\hat{\mathbf{x}}), \hat{\mathbf{o}}^i$, and $\hat{\mathbf{z}}^i(\hat{\mathbf{x}})$ are also functions of $\mathbf{x}$, which we omitted for simplicity. Since the new activation function $\hat{\mathbf{o}}$ is fixed given $\mathbf{x}$, effectively it applies the same linearity to $\hat{\mathbf{z}}_j^i$ as $\mathbf{z}_j^i$ in $S(\mathbf{x})$ and each $\hat{\mathbf{z}}_j^i(\hat{\mathbf{x}})$ is *linear* to $\hat{\mathbf{x}}, \forall \hat{\mathbf{x}} \in \mathbb{R}^D$. As a direct result of linearity and the equivalence of $g_\theta$ and $f_\theta$ (and all the respective $\hat{\mathbf{z}}_j^i$ and $\mathbf{z}_j^i$) in $S(\mathbf{x})$, we have to following equality:

$$\frac{\partial \hat{\mathbf{z}}_j^i(\hat{\mathbf{x}})}{\partial \hat{\mathbf{x}}_k} = \frac{\partial \mathbf{z}_j^i}{\partial \mathbf{x}_k}, \forall \hat{\mathbf{x}} \in \mathbb{R}^D. \tag{17}$$

We then do the following procedure to collect the partial derivatives with respect to an input axis $k$: 1) feed a zero vector $\mathbf{0}$ to $g_\theta$ to get $\hat{\mathbf{z}}_j^i(\mathbf{0})$ and 2) feed a unit vector $\mathbf{e}^k$ on the axis to get $\hat{\mathbf{z}}_j^i(\mathbf{e}^k)$. Then the derivative of each neuron $\mathbf{z}_j^i$ with respect to $\mathbf{x}_k$ can be computed as

$$\hat{\mathbf{z}}_j^i(\mathbf{e}^k) - \hat{\mathbf{z}}_j^i(\mathbf{0}) = \left[\nabla_{\mathbf{e}^k}\hat{\mathbf{z}}_j^i(\mathbf{e}^k)^\top \mathbf{e}^k + \hat{\mathbf{z}}_j^i(\mathbf{0})\right] - \hat{\mathbf{z}}_j^i(\mathbf{0}) = \left[\frac{\partial \hat{\mathbf{z}}_j^i(\mathbf{e}^k)}{\partial \mathbf{e}_k^k} \times 1 + \hat{\mathbf{z}}_j^i(\mathbf{0})\right] - \hat{\mathbf{z}}_j^i(\mathbf{0}) = \frac{\partial \mathbf{z}_j^i}{\partial \mathbf{x}_k}, \tag{18}$$

where the first equality comes from the linearity of $\hat{\mathbf{z}}_j^i(\hat{\mathbf{x}})$ with respect to any $\hat{\mathbf{x}}$. With the procedure, the derivative of *all the neurons* to an input dimension can be computed with 2 forward pass, which can be further scaled by computing all the gradients of $\mathbf{z}_j^i$ with respect to all the $D$ dimensions with $D + 1$ forward pass *in parallel*. We remark that the implementation is very simple and essentially the same across all the piecewise linear networks.

To analyze the complexity of the proposed approach, we assume that parallel computation does not incur any overhead and a batch matrix multiplication takes a unit operation. In this setting, a typical forward pass up to the last hidden layer takes $M$ operations. To compute the gradients of all the neurons for a batch of inputs, our perturbation algorithm first takes a forward pass to obtain the activation patterns for the batch of inputs, and then takes another forward pass with perturbations to obtain the gradients. Since both forward passes are done up to the last hidden layers, it takes $2M$ operations in total.

In contrast, back-propagation cannot be parallelized among neurons, so computing the gradients of all the neurons must be done sequentially. For each neuron $\mathbf{z}_j^i$, it takes $2i$ operations for back-propagation to compute its gradient ($i$ operations for each of the forward and backward pass). Hence, it takes $\sum_{i=1}^{M} 2iN_i$ operations in total for back-propagations to compute the same thing.

## D  DYNAMIC PROGRAMMING THE GRADIENTS

We can exploit the chain-rule of Jacobian to do dynamic programming for computing all the gradients of $\mathbf{z}_j^i$. Note that all the gradients of $\mathbf{z}_j^i$ in the $i^{\text{th}}$ layer can be represented by the Jacobian $J_{\mathbf{x}}\mathbf{z}^i$. Then 1) For the first layer, the Jacobian is trivially $J_{\mathbf{x}}\mathbf{z}^1 = \mathbf{W}^1$. 2) We then iterate higher layers with the Jacobian of previous layers by chain rules $J_{\mathbf{x}}\mathbf{z}^i = \mathbf{W}^i J_{\mathbf{z}^{i-1}}\mathbf{a}^{i-1} J_{\mathbf{x}}\mathbf{z}^{i-1}$, where $J_{\mathbf{x}}\mathbf{z}^{i-1}$ and $\mathbf{W}^i$ are stored and $J_{\mathbf{z}^{i-1}}\mathbf{a}^{i-1}$ is simply the Jacobian of activation function (a diagonal matrix with $0/1$ entries for ReLU activations). The dynamic programming approach is efficient for fully connected networks, but is inefficient for convolutional layers, where explicitly representing the convolutional operation in the form of linear transformation ($\in \mathbb{R}^{N_{i+1} \times N_i}$) is expensive.

## E  DERIVATIONS FOR MAXOUT/MAX-POOLING NONLINEARITY

Here we only make an introductory guide to the derivations for maxout/max-pooling nonlinearity. The goal is to highlight that it is feasible to derive inference and learning methods upon a piecewise

linear network with max-pooling nonlinearity, but we do not suggest to use it since a max-pooling neuron would induce new linear constraints; instead, we suggest to use convolution with large strides or average-pooling which do not incur any constraint.

For simplicity, we assume the target network has a single nonlinearity, which maps $N$ neurons to 1 output by the maximum

$$\mathbf{a}_1^1 = \max(\mathbf{z}_1^1, \ldots, \mathbf{z}_N^1), \mathbf{z}^1 = \mathbf{W}^1\mathbf{x} + \mathbf{b}^1. \tag{19}$$

Then we can define the corresponding activation pattern $\mathbf{o} = \mathbf{o}_1^1 \in [N]$ as which input is selected:

$$\mathbf{z}_i^1 \geq \mathbf{z}_j^1, \forall j \neq i \in [N], \ \ \text{if } \mathbf{o}_1^1 = i. \tag{20}$$

It is clear to see once an activation pattern is fixed, the network again degenerates to a linear model, as the nonlinearity in the max-pooling effectively disappears. Such activation pattern induces a feasible set in the input space where derivatives are guaranteed to be stable, but such representation may have a similar degenerate case where two activation patterns yield the same linear coefficients.

The feasible set $S(\mathbf{x})$ of a feasible activation pattern $\mathcal{O} = \{\mathbf{o}_1^1\}$ at $\mathbf{x}$ can be derived as:

$$S(\mathbf{x}) = \cap_{j \in [N] \setminus \{i\}} \{\bar{\mathbf{x}} \in \mathbb{R}^D : (\nabla_{\mathbf{x}}\mathbf{z}_i^1)^\top\bar{\mathbf{x}} + (\mathbf{z}_i^1 - (\nabla_{\mathbf{x}}\mathbf{z}_i^1)^\top\mathbf{x}) \geq (\nabla_{\mathbf{x}}\mathbf{z}_j^1)^\top\bar{\mathbf{x}} + (\mathbf{z}_j^1 - (\nabla_{\mathbf{x}}\mathbf{z}_j^1)^\top\mathbf{x})\}. \tag{21}$$

To check its correctness, we know that Eq. (21) is equivalent to

$$S(\mathbf{x}) = \cap_{j \in [N] \setminus \{i\}} \{\bar{\mathbf{x}} \in \mathbb{R}^D : \mathbf{W}_i^1\bar{\mathbf{x}} + \mathbf{b}_i^1 \geq \mathbf{W}_j^1\bar{\mathbf{x}} + \mathbf{b}_j^1\} \tag{22}$$

$$= \cap_{j \in [N] \setminus \{i\}} \{\bar{\mathbf{x}} \in \mathbb{R}^D : (\mathbf{W}_i^1 - \mathbf{W}_j^1)\bar{\mathbf{x}} + (\mathbf{b}_i^1 - \mathbf{b}_j^1) \geq 0\}, \tag{23}$$

where the linear constraints are evident, and the feasible set is thus again a convex polyhedron. As a result, all the inference and learning algorithms can be applied with the linear constraints. Clearly, for each max-pooling neuron with $N$ inputs, it will induce $N - 1$ linear constraints.

## F    IMPLEMENTATION DETAILS ON THE TOY DATASET

The FC model consists of $M = 4$ fully-connected hidden layers, where each hidden layer has 100 neurons. The input dimension $D$ is 2 and the output dimension $L$ is 1. The loss function $\mathcal{L}(f_\theta(\mathbf{x}), \mathbf{y})$ is sigmoid cross entropy. We train the model for 5000 epochs with Adam (Kingma & Ba, 2015) optimizer, and select the model among epochs based on the training loss. We fix $C = 5$, and increase $\lambda \in \{10^{-2}, \ldots, 10^2\}$ for both the distance regularization and relaxed regularization problems until the resulting classifier is not perfect. The tuned $\lambda$ in both cases are 1.

## G    IMPLEMENTATION DETAILS ON MNIST DATASET

The data are normalized with $\mu = 0.1307$ and $\sigma = 0.3081$. We first compute the margin $\hat{\epsilon}_{\mathbf{x},p}$ in the normalized data, and report the scaled margin $\sigma\hat{\epsilon}_{\mathbf{x},p}$ in the table, which reflects the actual margin in the original data space since

$$\|\mathbf{x} - \mathbf{x}'\|_p = \sigma\|\mathbf{x}/\sigma - \mathbf{x}'/\sigma\|_p = \sigma\|(\mathbf{x} - \mu)/\sigma - (\mathbf{x}' - \mu)/\sigma\|_p, \tag{24}$$

so the reported margin should be perceived in the data space of $\mathcal{X} = [0, 1]^{28 \times 28}$.

We compute the exact ROLL loss during training (i.e., approximate learning is not used). The FC model consists of $M = 4$ fully-connected hidden layers, where each hidden layer has 300 neurons. The activation function is ReLU. The loss function $\mathcal{L}(f_\theta(\mathbf{x}), \mathbf{y})$ is a cross-entropy loss with soft-max performed on $f_\theta(\mathbf{x})$. The number of epochs is 20, and the model is chosen from the best validation loss from all the epochs. We use stochastic gradient descent with Nesterov momentum. The learning rate is 0.01, the momentum is 0.5, and the batch size is 64.

**Tuning:** We do a grid search on $\lambda, C, \gamma$, with $\lambda \in \{2^{-3}, \ldots, 2^2\}$, $C \in \{2^{-2}, \ldots, 2^3\}$, $\gamma \in \{\max, 25, 50, 75, 100\}$ (max refers to Eq. (7)), and report the models with the largest validation $\hat{\epsilon}_{2,50}$ given the same and 1% less validation accuracy compared to the baseline model (the vanilla loss).

## H  IMPLEMENTATION DETAILS ON THE JAPANESE VOWEL DATASET

The data are not normalized.

We compute the exact ROLL loss during training (i.e., approximate learning is not used). The representation is learned with a single layer scoRNN, where the state embedding from the last timestamp for each sequence is treated as the representation along with a fully-connected layer to produce a prediction as $f_\theta(\mathbf{x})$. We use LeakyReLU as the activation functions in scoRNN. The dimension of hidden neurons in scoRNN is set to 512. The loss function $\mathcal{L}(f_\theta(\mathbf{x}), \mathbf{y})$ is a cross-entropy loss with soft-max performed on $f_\theta(\mathbf{x})$. We use AMSGrad optimizer (Reddi et al., 2018). The learning rate is 0.001, and the batch size is 32 (sequences).

**Tuning:** We do a grid search on $\lambda \in \{2^{-6}, \ldots, 2^3\}$, $C \in \{2^{-5}, \ldots, 2^7\}$, and set $\gamma = 100$. The models with the largest testing $\hat{\epsilon}_{2,50}$ given the same and 1% less testing accuracy compared to the baseline model (the vanilla loss) are reported. (We do not have validation data in this dataset, so the performance should be interpreted as validation.)

## I  IMPLEMENTATION DETAILS ON CALTECH-256 DATASET

The data are normalized with

$$\mu = [0.485, 0.456, 0.406], \text{ and } \sigma = [0.225, 0.225, 0.225] \tag{25}$$

along each channel. We train models on the normalized images, and establish a bijective mapping between the normalized distance and the distance in the original space with the trick introduced in Appendix G. The bijection is applied to our sample-based approach to compute $\mathbb{E}_{\mathbf{x}' \sim \text{Unif}(\mathcal{B}_{8/256,\infty}(\mathbf{x}) \cap \mathcal{X})}[\Delta(\mathbf{x}, \mathbf{x}', \mathbf{y})]$ and $\max_{\mathbf{x}' \in \mathcal{B}_{8/256,\infty}(\mathbf{x}) \cap \mathcal{X}}[\Delta(\mathbf{x}, \mathbf{x}', \mathbf{y})]$ that we ensure the perturbed space is consistent with $\mathcal{X} = [0, 1]^{299 \times 299 \times 3}$ and $\mathcal{B}_{8/256,\infty}(\mathbf{x})$ in the original space.

We download the pre-trained ResNet-18 (He et al., 2016) from PyTorch (Paszke et al., 2017), and we revise the model architecture as follows: 1) we replace the max-pooling after the first convolutional layer with average-pooling to reduce the number of linear constraints (because max-pooling induces additional linear constraints on activation pattern, while average-pooling does not), and 2) we enlarge the receptive field of the last pooling layer such that the output will be 512 dimension, since ResNet-18 is originally used for smaller images in ImageNet data (most implementations use $224 \times 224 \times 3$ dimensional images for ImageNet while our data has even higher dimension $299 \times 299 \times 3$).

We train the model with stochastic gradient descent with Nesterov momentum for 20 epochs. The initial learning rate is 0.005, which is adjusted to 0.0005 after the first 10 epochs. The momentum is 0.5. The batch size is 32. The model achieving the best validation loss among the 20 epochs is selected.

**Tuning:** Since the training is computationally demanding, we first fix $C = 8$, use only 18 samples (6 per channel) for approximate learning, and tune $\lambda \in \{10^{-6}, 10^{-5}, \ldots\}$ until the model yields significantly inferior validation accuracy than the vanilla model. Afterwards, we fix $\lambda$ to the highest plausible value ($\lambda = 0.001$) and try to increase $C \in \{8, 80, \ldots\}$, but we found that $C = 8$ is already the highest plausible value. Finally, we train a model with 360 random samples (120 per channel) for approximate learning to improve the quality of approximation.

## J  IMPLEMENTATION DETAILS OF THE GENETIC ALGORITHM

We implement a genetic algorithm (GA) (Whitley, 1994) with 4800 populations $\mathbf{P}$ and 30 epochs. Initially, we first uniformly sample 4800 samples (called chromosome in GA literature) in the domain $\mathcal{B}_{\epsilon,\infty}(\mathbf{x}) \cap \mathcal{X}$ for $\mathbf{P}$. In each epoch,

1. $\forall \mathbf{c} \in \mathbf{P}$, we evaluate the $\ell_1$ distance of its gradient from that of the target $\mathbf{x}$:

$$\|\nabla_\mathbf{c} f_\theta(\mathbf{c})_\mathbf{y} - \nabla_\mathbf{x} f_\theta(\mathbf{x})_\mathbf{y}\|_1 \tag{26}$$

2. (Selection) we sort the samples based on the $\ell_1$ distance and keep the top $25\%$ samples in the population (denoted as $\hat{\mathbf{P}}$).

3. (Crossover) we replace the remaining $75\%$ samples with a random linear combination of a pair $(\mathbf{c}, \mathbf{c}')$ from $\hat{\mathbf{P}}$ as:

$$\bar{\mathbf{c}} = \alpha\mathbf{c} + (1 - \alpha)\mathbf{c}', \mathbf{c}, \mathbf{c}' \in \hat{\mathbf{P}}, \alpha \in \text{Unif}([-0.25, 1.25]). \tag{27}$$

4. (Projection) For all the updated samples $\mathbf{c} \in \mathbf{P}$, we do an $\ell_\infty$-projection to the domain $\mathcal{B}_{\epsilon,\infty}(\mathbf{x}) \cap \mathcal{X}$ to ensure the feasibility.

Finally, the sample in $\mathbf{P}$ that achieves the maximum $\ell_1$ distance is returned. We didn't implement mutation in our GA algorithm due to computational reasons. For the readers who are not familiar with GA, we comment that the crossover operator is analogous to a gradient step where the direction is determined by other samples and the step size is determined randomly.

## K  VISUALIZATION OF ADVERSARIAL GRADIENTS IN CALTECH-256 DATASET

We visualize the following images:

- Original image.
- Original gradient: the gradient on the original image.
- Adversarial gradient: the maximum $\ell_1$ distorted gradient in $\mathcal{B}_{\epsilon,\infty}(\mathbf{x}) \cap \mathcal{X}$.
- Image of adv. gradient: the image that yields adversarial gradient.
- Original int. gradient: the integrated gradient attribution (Sundararajan et al., 2017) on the original image.
- Adversarial int. gradient: the integrated gradient attribution (Sundararajan et al., 2017) on the 'image of adv. gradient'. Note that we didn't perform optimization to find the image that yields the maximum distorted integrated gradient.

We follow a common implementation in the literature (Smilkov et al., 2017; Sundararajan et al., 2017) to visualize gradients and integrated gradients by the following procedure:

1. Aggregating derivatives in each channel by summation.
2. Taking absolute value of aggregated derivatives.
3. Normalizing the aggregated derivatives by the $99^{\text{th}}$ percentile
4. Clipping all the values above $1$.

After this, the derivatives are in the range $[0, 1]^{299 \times 299}$, which can be visualized as a gray-scaled image. The original integrated gradient paper visualizes the element-wise product between the gray-scaled integrated gradient and the original image, but we only visualize the integrated gradient to highlight its difference in different settings since the underlying images (the inputs) are visually indistinguishable.

We visualize the examples in Caltech-256 dataset that yield the $P_{25}, P_{50}, P_{75}, P_{100}$ ($P_r$ denotes the $r^{\text{th}}$ percentile) of the maximum $\ell_1$ gradient distortions among the testing data on our ROLL model in Figure 5 and 6, where the captions show the exact values of the maximum $\ell_1$ gradient distortion for each image. Note that the exact values are slightly different from Table 4, because each percentile in Table 4 is computed by an interpolation between the closest ranks (as in `numpy.percentile`), and the figures in Figure 5 and 6 are chosen from the images that are the closest to the percentiles.

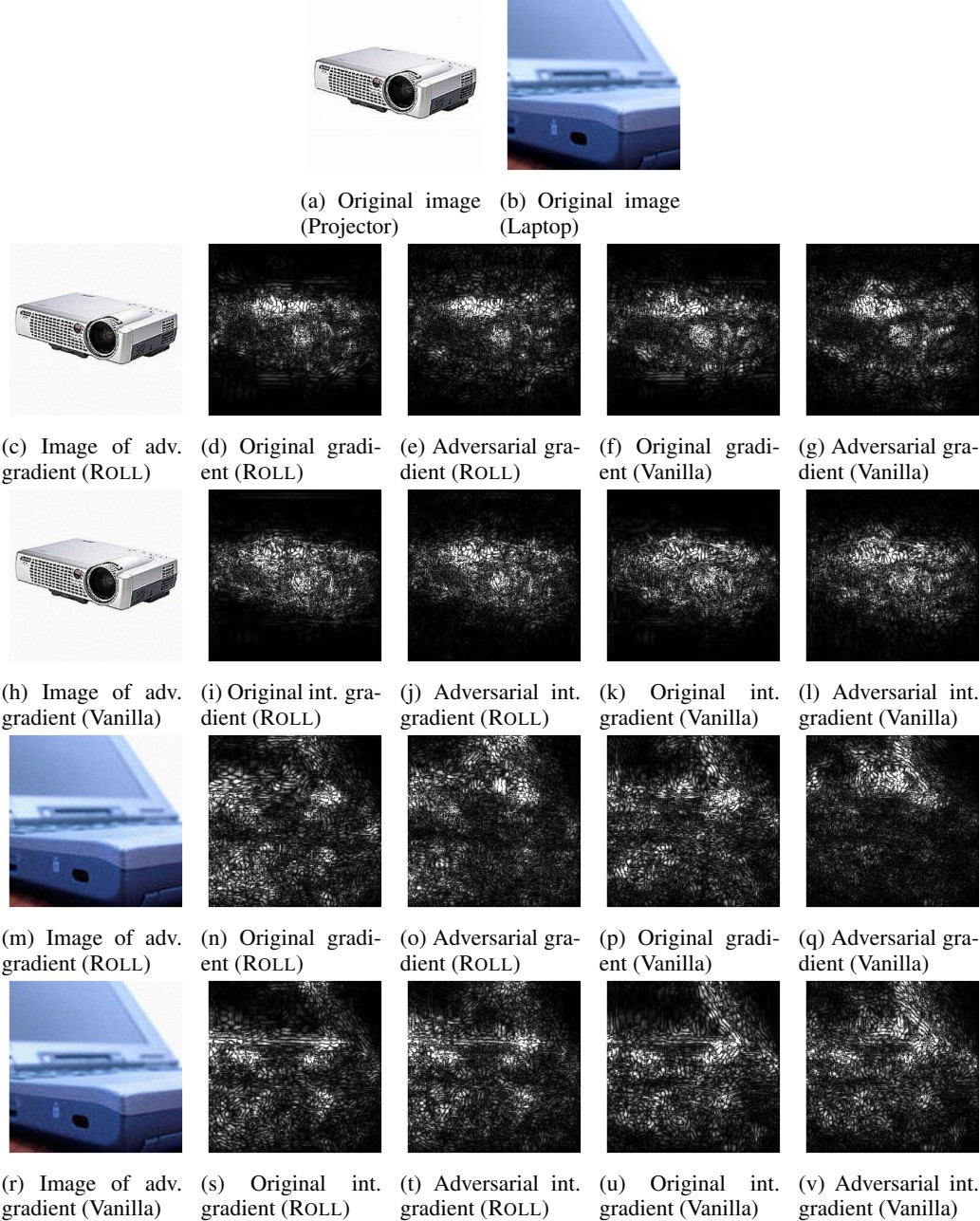

Figure 5: Visualization of the examples in Caltech-256 dataset that yield the $P_{25}$ (above) and $P_{50}$ (below) of the maximum $\ell_1$ gradient distortions among the testing data on our ROLL model. For the vanilla model, the maximum $\ell_1$ gradient distortion $\Delta(\mathbf{x}, \mathbf{x}', \mathbf{y})$ is equal to 893.3 for 'Projector' in Figure 5g and 1199.4 for 'Laptop' in Figure 5q. For the ROLL model, the maximum $\ell_1$ gradient distortion $\Delta(\mathbf{x}, \mathbf{x}', \mathbf{y})$ is equal to 779.9 for 'Projector' in Figure 5e and 1045.4 for 'Laptop' in Figure 5o.

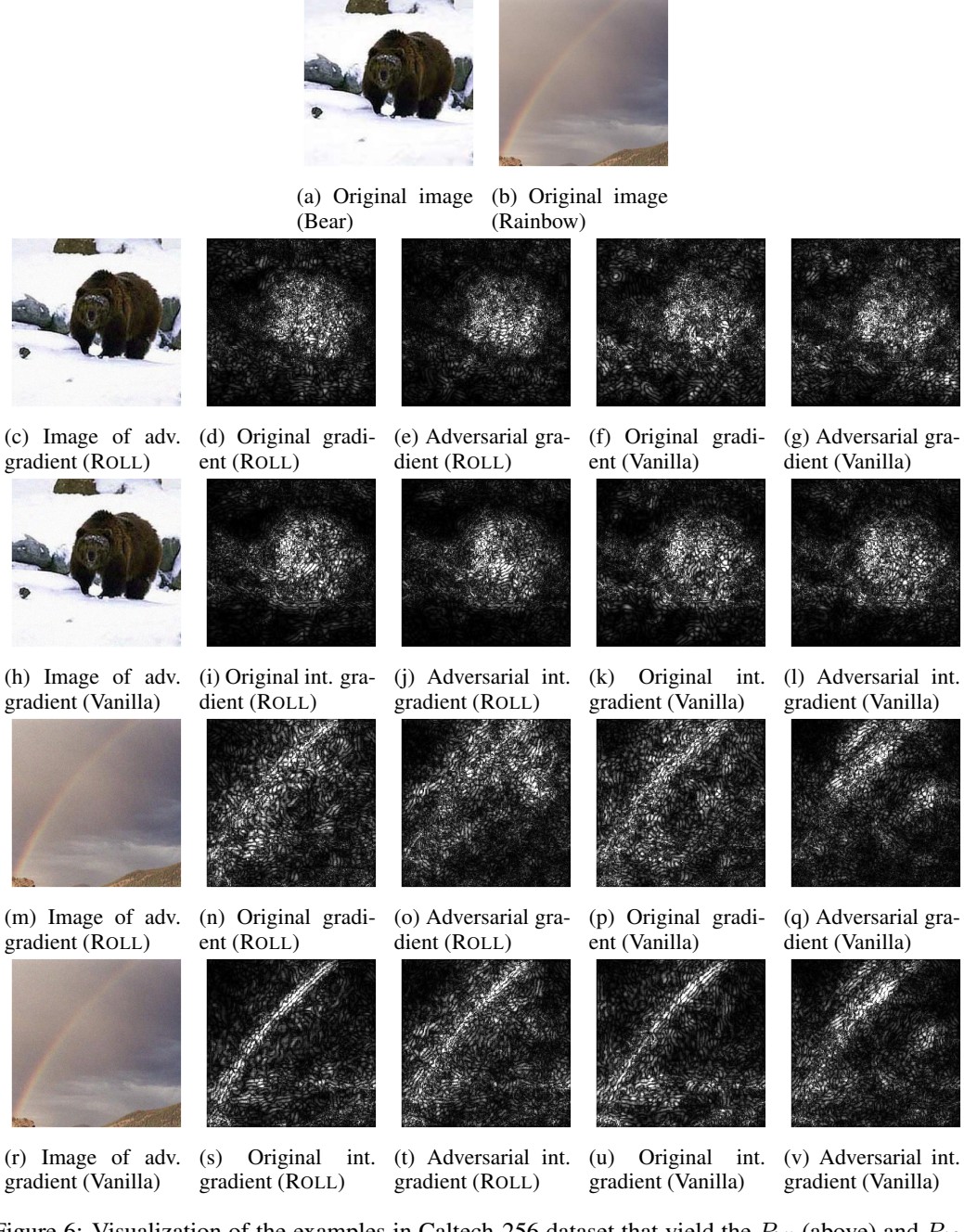

Figure 6: Visualization of the examples in Caltech-256 dataset that yield the $P_{75}$ (above) and $P_{100}$ (below) of the maximum $\ell_1$ gradient distortions among the testing data on our ROLL model. For the vanilla model, the maximum $\ell_1$ gradient distortion $\Delta(\mathbf{x}, \mathbf{x}', \mathbf{y})$ is equal to $1547.1$ for 'Bear' in Figure 6g and $5473.5$ for 'Rainbow' in Figure 6q. For the ROLL model, the maximum $\ell_1$ gradient distortion $\Delta(\mathbf{x}, \mathbf{x}', \mathbf{y})$ is equal to $1367.9$ for 'Bear' in Figure 6e and $3882.8$ for 'Rainbow' in Figure 6o.

