# OpenReview forum: "Towards Robust, Locally Linear Deep Networks"
_ICLR.cc/2019/Conference_

### Official Review · AnonReviewer1 · 2018-10-29
**This compelling theoretical framework could benefit from more applications.**

**Rating:** 7
**Confidence:** 4

**Review:**

The paper considers deep nets with piecewise linear activation functions, which are known to give rise to piecewise linear input-output mappings, and proposes loss functions which discourage datapoints in the input space from lying near the boundary between linear regions. These loss functions are well-motivated theoretically, and have the intended effect of increasing the distance to the nearest boundary and reducing the number of distinct linear regions.

My only concern is that while their method appears to effectively increase the l_1 and l_2 margin (as they have defined it), the utility of doing so is not clearly demonstrated. If improving the quality or validity of local linearization for explaining predictions is one of the main motivations for this work, showing that the proposed method does so would strengthen the overall message. However, I do feel that “establishing robust derivatives over larger regions” is an important problem in its own right.

With the exception of some minor typos, the exposition is clear and the theoretical claims all appear correct. The authors may have missed some relevant recent work [1], but their contributions are complementary. It is not immediately clear that the parallel computation of gradients proposed in section 4.1 is any faster than standard backpropagation, as this has to be carried out separately for each linear region. A basic complexity analysis or running time comparison would help clarify this. I think I am missing the point of the gradient visualizations in figure 4, panels b-e and g-j.


[1] Elsayed, Gamaleldin F., et al. "Large Margin Deep Networks for Classification." arXiv preprint arXiv:1803.05598 (To appear in NIPS 2018).

---

> ### Author Response · Authors · 2018-11-13
> **Thank you for the comments. Please note that general responses are provided above.**
>
> Q: missing relevant recent work
>
> Thanks for pointing out this work. We will read it and then add it to our related work.

---

### Official Review · AnonReviewer2 · 2018-11-01
**It seems to be an interesting problem but is the solution proposed practical?**

**Rating:** 8
**Confidence:** 3

**Review:**


########## Updated Review ##########

The author(s) have presented a very good rebuttal, and I am impressed. My concerns have been addressed and my confusions have been clarified. To reflect this, I am raising my points to 8. It is a good paper, job well done. I enthusiastically recommend acceptance.

################################

A key challenge that presents the deep learning community is that state-of-the-art solutions are oftentimes associated with unstable derivatives, compromising the robustness of the network. In this paper, the author(s) explore the problem of how to train a neural network with stable derivatives by expanding the linear region associated with training samples.

The author(s) studied deep networks with piecewise linear activations, which allow them to derive lower bounds on the $l_p$ margin with provably stable derivatives. In the special case of $l_2$ metric, this bound is analytic, albeit rigid and non-smooth. To avoid associated computational issues, the author(s) borrowed an idea from transductive/semi-supervised SVM (TSVM) to derive a relaxed formulation.

In general, I find this paper rather interesting and well written. However, I do have a few concerns and confusions as listed below:

Major ones:

- I would prefer some elaborations on why the relaxation proposed in Eqn (8) serves to encourage the margin of L2 ball? What's the working mechanism or heuristic behind this relaxation? This is supposedly one of the key techniques used in optimization, yet remains obscure.

- On empirical gains, the author(s) claimed that "about 30 times larger margins can be achieved by trading off 1% accuracy."  It seems that consistently yields inferior prediction accuracy. In my opinion, the author(s) failed to showcase the practical utility of their solution. A better job should be done to validate the claim `` The problem we tackle has implications to interpretability and transparency of complex models. ''

- As always, gradient-based penalties suffer from heavy computational overhead. The final objectives derived in this paper (Eqn (7) & Eqn (9)) seem no exception to this, and perhaps even worse since the gradient is taken wrt each neuron. Could the author(s) provide statistics on empirical wallclock performance? How much drain does this extra gradient penalty impose on the training efficiency?

Minor ones:

- Just to clarify, does the | - | used in Eqn (9) for |I(x,\gamma)|  denote counting measure?

- I do not see the necessity of introducing Lemma 7 in the text. Please explain.

- Lemma 8, ``... then any optimal solutions for the problem is also optimal for Eqn (4). '' Do you mean ``the following problem'' (Eqn (5))?

---

> ### Author Response · Authors · 2018-11-13
> **Thank you for the comments. Please note that general responses are provided above.**
>
> Major 1: I would prefer some elaborations on why the relaxation proposed in Eqn (8) serves to encourage the margin of L2 ball? What's the working mechanism or heuristic behind this relaxation? This is supposedly one of the key techniques used in optimization, yet remains obscure.
>
> We will add more explanation to make it clearer. The working mechanism is based on the theoretical bounds and Lagrangian relaxations. Briefly, the derivation proceeds in two parts. In the first part, Lemma 8 (Eq. 5) simply rewrites Eq. (4) in a constraint form but needs to assume that a non-zero margin exists. To get Eq. (6), we use the fact that now |z^i_j|>=1 and thus the numerator in the margin in Eq. (5) can be lower bounded by 1, implying an upper bound on the overall learning objective. In the second part, we note that Eq. (6) is now akin to a hard-margin SVM/TSVM (see [1]). The constraint can be relaxed to a Lagrangian form resulting in Eq. (7) and the TSVM can be analogously relaxed to Eq. (8). To see the correspondences, note that in a single neuron case the gradient \nabla_x z^i_j and z^i_j in Eq. (7) simply correspond to w and w^T x + b in Eq. (8).
>
> [1] Boser, Bernhard E., Isabelle M. Guyon, and Vladimir N. Vapnik. "A training algorithm for optimal margin classifiers." Proceedings of the fifth annual workshop on Computational learning theory. ACM, 1992.
>
> Major 2 - 1: On empirical gains, the author(s) claimed that "about 30 times larger margins can be achieved by trading off 1% accuracy."  It seems that consistently yields inferior prediction accuracy.
>
> The performance might indeed degrade if one really pursues an extremely large linear region (e.g., 30 times larger than a vanilla model). However, in Table 1 and 2, we also show that for reasonable parameter choices, our loss can achieve the same accuracy with more robust derivatives. For example, in MNIST, our approach exhibits 10 times larger locally linear regions given the same accuracy. For the ResNet experiment in Table 3, our approach even improves the accuracy.
>
> The more important message we want to convey is that in some cases when robustness of derivatives is a requirement (e.g., a robust explanation for a sensitive decision), we provide a way to set the trade-off.
>
> Major 2 - 3: A better job should be done to validate the claim `` The problem we tackle has implications to interpretability and transparency of complex models. ''
>
> While this has been partially answered in the general comments, it's an important point, so we provide a more detailed answer here. Our claim of implication to transparency is supported by our results on stability of gradient-based explanations (Section 5.3).
>
> The gradient saliency map is a well-known interpretability method for deep models. Our inference solution verifies the $l_p$ margins where such interpretation is guaranteed to be stable, and our learning algorithm stabilizes the explanations as validated through the $l_p$ margins and gradient distortions.
>
> Minor 1: Just to clarify, does the | - | used in Eqn (9) for |I(x,\gamma)|  denote counting measure?
>
> Yes, this is correct. We will add a description below Eq. (9) to clarify it. Thank you for the comment.
>
> Minor 2: I do not see the necessity of introducing Lemma 7 in the text. Please explain.
>
> Thank you for the question, we have updated the paper to address it.
>
> Lemma 7 is used in Table 1 to compute the number of complete linear regions (#CLR). As mentioned in the last paragraph of Section 5.1, “The lower #CLR in our approach than the baseline model reflects the existence of certain larger linear regions that span across different testing points”, so it serves as an indirect measurement for the size of linear regions.
>
> Minor 3: Lemma 8, ``... then any optimal solutions for the problem is also optimal for Eqn (4). '' Do you mean ``the following problem'' (Eqn (5))?
>
> Yes, it is correct. We have updated the paper to address it.

---

### Official Review · AnonReviewer3 · 2018-11-02
**Very nice work with clear intuition and impressive results**

**Rating:** 8
**Confidence:** 4

**Review:**

1. This is a very relevant and timely work related to robustness of deep learning models under adversarial attacks.

2. In recent literature of verifiable/certifiable networks, (linear) ReLU network has emerged as a tractable model architecture where analytically sound algorithms/understanding can be achieved. This paper adopts the same setting, but very clearly articulates the differences between this work and the other recent works (Weng et al 2018, Wong et al. 2018).

3. The primary innovation here is that the authors not only identify the locally linear regions in the loss surface but expand that region by learning essentially leading to gradient stability.

4. A very interesting observation is that the robustifying process does not really reduce the overall accuracy which is the case of many other methods.

5. The visualizations show the stability properties nicely, but a bit more explanations of those figures would help the readers quite a bit.

6. While I understand some of the feasibility issues associated with other existing methods, it would be interesting to try to compare performance (if not exact performance, the at least loss/gradient surfaces etc.) with some of them.

7. The adversarial scenarios need to be explained better.

---

> ### Author Response · Authors · 2018-11-13
> **Thank you for the comments. Please note that general responses are provided above.**
>
> Q5. The visualizations show the stability properties nicely, but a bit more explanations of those figures would help the readers quite a bit.
>
> Thanks for the comment. We can certainly add more explanations about the figures. Is there a specific figure you were referring to?
>
> Q6. While I understand some of the feasibility issues associated with other existing methods, it would be interesting to try to compare performance (if not exact performance, the at least loss/gradient surfaces etc.) with some of them.
>
> We are not aware of any directly comparable existing method for establishing robust derivatives, so we focus on an ablation setting comparing a vanilla loss with the proposed robust loss in various circumstances (FC networks, RNN, and ResNet). Existing methods using activation patterns (e.g., adversarial defense in (Wong & Kolter, 2018)) are not directly comparable to our work. We expand regions where gradients are invariant. However, gradients can be large or small even if invariant over a larger region. In contrast, any large gradient will likely lead to an adversarial example.

---

### Public Comment · (anonymous) · 2018-11-07
**Comparison to prior scalable provability training**

This seems like an interesting result but I am curious how it compares with other scalable verification and training techniques such as those proposed in Mirman et al's "Differentiable Abstract Interpretation for Provably Robust Neural Networks.", ICML'18 and Croce et al's "
Provable Robustness of ReLU networks via Maximization of Linear Regions" and Gowal et al's "On the Effectiveness of Interval Bound Propagation for Training Verifiably Robust Models"?

---

> ### Author Response · Authors · 2018-11-26
> **Official response**
>
> Thank you for asking. As elaborated in the introduction (3rd paragraph), our goal of establishing gradient stability is different from adversarial learning (e.g., output stability as in the referred papers).
>
> Technically, our approach is indeed relevant to the solution of (Croce et al.). We will be happy to discuss and cite the paper in the camera ready version. However, please note that (Croce et al.) was posted after the ICLR submission deadline, so it cannot be viewed as prior work (it’s the other way around).

---

### Author Response · Authors · 2018-11-13
**General Response - 1 & 2**

We thank all the reviewers for the insightful comments, suggestions and questions. The general responses are provided here, while specific questions are responded individually. Here we focus on the utility of our approach, implications to interpretability, as well as complexity of the perturbation algorithm.

1. The utility of doing so is not clearly demonstrated. (R1) / In my opinion, the author(s) failed to showcase the practical utility of their solution. (R2)

We believe establishing robust derivatives is important in its own right: stable derivatives serve many roles, including interpretability, but require extra effort to achieve in deep models. Note that robustness of explanations is an open problem in the community that has received significant interest over the past year (Ghorbani et al., 2017; Alvarez-Melis & Jaakkola, 2018a). Given this premise, rather than showcasing the utility of gradient stability, we focus on showing that our method yields more robust derivatives across many architectures by measuring margins/gradient distortions. We also included an application of inducing robust explanations in gradient saliency maps.

2. If improving the quality or validity of local linearization for explaining predictions is one of the main motivations for this work, showing that the proposed method does so would strengthen the overall message. / The point of gradient visualization in Figure 4 is not clear. (R1) / A better job should be done to validate the claim `` The problem we tackle has implications to interpretability and transparency of complex models. '' (R2) / The adversarial scenarios need to be explained better. (R3)

The goal of the adversarial scenario is to analyze the robustness of gradient explanations either coming from a typical deep model or after our method. We do so by visualizing how the gradient can change (i.e., be distorted) in the worst case in a small neighborhood. In Figure 4, we show that our approach yields more robust gradient saliency maps than a vanilla deep model. Note that the margin analyses in the experiments section yield the same conclusion, since gradients within any established margin remain the same. The proposed inference (certifying the margin) and learning (expanding the margin) algorithms thus directly contribute to interpretability. Indeed, an unstable explanation of a critical decision (e.g., medical/financial/security decisions) would likely be unacceptable. We certify and enlarge the “margin of validity” of such explanations.

---

### Author Response · Authors · 2018-11-13
**General Response - 3**

3. Basic complexity analysis or running time comparison for gradient computation. (R1) / The gradient-based penalties suffer from heavy computational overhead. How much drain does this extra gradient penalty impose on the training efficiency. (R2)

Condensed versions of the complexity analysis and empirical running time have been added to the paper.

I. complexity

We assume that (1) parallel computation does not incur any overhead, and (2) batch matrix multiplication takes a unit operation. The notation is consistent with the paper in that M refers to the number of hidden layers and N_i refers to the number of neurons in the i^th layer.

In short, for a batch of samples,
0. It takes M operations for a forward pass up to the last hidden layer.
1. our perturbation algorithm take 2M operations to compute the gradients of all the neurons.
2. Straightforward back-propagation takes \sum_{i=1}^M [ 2i x N_i ] operations to compute the gradients of all the neurons. Note that vanilla back-propagation requires (# of neurons) sequential calls, since it cannot be parallelized across neurons for this type of gradients. The details are in the updated paper draft.

The proposed approach is then, to our best knowledge, the first algorithm that is architecture-agnostic and has a tractable complexity to compute the gradient for all the neurons.

II. running time

We measure the running time for the 4-layer FC networks on MNIST. To accurately analyze the difference, we report the running time for performing a complete mini-batch gradient descent step (from the initial forward pass to the final gradient update) in each iteration:

It takes
1. Vanilla loss: 						                                0.00129 sec
2. Full ROLL loss computed by back-propagation: 		0.31185 sec
3. Full ROLL loss computed by perturbation: 			0.02667 sec
4. Approximate ROLL loss computed by perturbation: 	0.00298 sec
on average for a complete mini-batch gradient descent update.

The full ROLL loss refers to Eq. (9) (gamma=100), and the approximate ROLL loss refers to Eq. (10), where we use 3 samples (1 / 256 input dimensions) to approximate the ROLL loss. The accuracy / median l_2 margin of the approximate ROLL loss (0.9782 / 0.0094) is comparable to the full ROLL loss (0.9761 / 0.0092). In total, it takes the vanilla loss 42.49 seconds and the approximate ROLL loss 69.94 seconds to complete training for 20 epochs. Overall, our approach is 2.3 times slower in gradient update, and only 1.6 times slower in total than the vanilla loss. The approximate loss is about 9 times faster than the full loss. Compared to back-propagation, our perturbation algorithm achieves about 12 times empirical speed-up. In summary, the computational overhead of our method is minimal compared to vanilla training, which is achieved by the perturbation algorithm and the approximate loss.

Note that the full and approximate ROLL losses actually have the same number of operations (in parallel) but their running times are different because parallel batch matrix multiplication does not take constant time for different batch sizes in practice. If we simply use the approximate version to avoid the overhead for large batches, the empirical running time indeed matches our complexity analysis (roughly 2M for the ROLL loss versus M for the vanilla loss).

For training ResNet on Caltech-256, when the sub-samples can fit the GPU memory, it takes less than 1 day to complete training for the approximate ROLL loss. Note that all of our experiments are done on single TITAN X GPU with 12G memory.

---

### Public Comment · (anonymous) · 2018-11-24
**Lack of experiments on CIFAR10 or CIFAR100 or ImageNet**

Why didn't you conduct any experiments on CIFAR10 or  CIFAR100 or ImageNet, and test the acc and #CLR as well as the margin ? The experiment on the toy dataset MNIST is not convincing.

---

> ### Author Response · Authors · 2018-11-26
> **Official response**
>
> The manageable size of MNIST was beneficial for parameter analysis and to illustrate properties of the method, while the more challenging Caltech-256 (299x299x3 dimensions) with ResNet was used to demonstrate scalability. Our method is not limited nor specifically tailored to image classification (cf. our sequence dataset example). While additional experiments (e.g., using the referred datasets) would always help, we did not see it as necessary.

---

### Public Comment · (anonymous) · 2018-12-20
**Missing relevant work**

Thanks for the enlightening paper. I believe there is one missed relevant work: "Deep Defense: Training DNNs with Improved Adversarial Robustness (arXiv:1803.00404, NeurIPS 2018)" which also aims at enlarging the l_p margin.

---

### Meta-Review · Area_Chair1 · 2018-12-12
**Novel work, and potentially of broader interest**

**Confidence:** 5
**Recommendation:** Accept (Poster)

**Metareview:**

The paper aims to encourage deep networks to have stable derivatives over larger regions under networks with piecewise linear activation functions.

All reviewers and AC note the significance of the paper. AC also thinks this is also a very timely work and potentially of broader interest of ICLR audience.